# Prefrontal-amygdalar oscillations related to social behavior in mice

Nahoko Kuga[1,2], Reimi Abe[1], Kotomi Takano[3], Yuji Ikegaya[1,4,5], Takuya Sasaki[1,2]*

[1]Laboratory of Chemical Pharmacology, Graduate School of Pharmaceutical Sciences, The University of Tokyo, Tokyo, Japan; [2]Department of Pharmacology, Graduate School of Pharmaceutical Sciences, Tohoku University, Sendai, Japan; [3]School of Medicine, Hiroshima University, Hiroshima, Japan; [4]Institute for AI and Beyond, The University of Tokyo, Tokyo, Japan; [5]Center for Information and Neural Networks, National Institute of Information and Communications Technology, Osaka, Japan

**Abstract** The medial prefrontal cortex and amygdala are involved in the regulation of social behavior and associated with psychiatric diseases but their detailed neurophysiological mechanisms at a network level remain unclear. We recorded local field potentials (LFPs) from the dorsal medial prefrontal cortex (dmPFC) and basolateral amygdala (BLA) while male mice engaged on social behavior. We found that in wild-type mice, both the dmPFC and BLA increased 4–7 Hz oscillation power and decreased 30–60 Hz power when they needed to attend to another target mouse. In mouse models with reduced social interactions, dmPFC 4–7 Hz power further increased especially when they exhibited social avoidance behavior. In contrast, dmPFC and BLA decreased 4–7 Hz power when wild-type mice socially approached a target mouse. Frequency-specific optogenetic manipulations replicating social approach-related LFP patterns restored social interaction behavior in socially deficient mice. These results demonstrate a neurophysiological substrate of the prefrontal cortex and amygdala related to social behavior and provide a unified pathophysiological understanding of neuronal population dynamics underlying social behavioral deficits.

*For correspondence:
t.sasaki.0224@gmail.com

Competing interest: The authors declare that no competing interests exist.

## Editor's evaluation

This manuscript is of broad interest to readers studying the neural basis of sociability, social anxiety, and anxiety-like behaviors. The authors use mouse models to understand how electrical oscillations in two key parts of the brain (prefrontal cortex and amygdala) relate to social behavior.

## Introduction

The medial prefrontal cortex (mPFC) plays a central role in social behavior (*Duncan and Owen, 2000*; *Wood and Grafman, 2003*; *Bicks et al., 2015*) through functional interactions with the amygdala (*Kumar et al., 2014*; *Adhikari et al., 2015*; *Bukalo et al., 2015*; *Tovote et al., 2015*), a region that is reciprocally connected with the mPFC (*Vertes, 2004*; *Hoover and Vertes, 2007*; *Adhikari et al., 2015*) and plays central roles in emotional responses such as fear and anxiety. A number of gene expression patterns and intracellular signaling pathways related to social behavior have been identified in the mPFC (*Bicks et al., 2015*; *Schubert et al., 2015*; *Yan and Rein, 2022*). Under pathological conditions, a number of studies have reported alterations in overall mPFC neuronal excitability and disruptions of mPFC-amygdala interactions in humans with psychiatric disorders with social behavior deficits such as autism spectrum disorders (ASD) and depression (*Happé et al., 1996*; *Castelli et al., 2002*; *Pierce et al., 2004*; *Greicius et al., 2007*; *Drysdale et al., 2017*) and animal models of these disorders (*Brumback et al., 2018*; *Abe et al., 2019*). A fundamental issue is how such molecular

and cellular mechanisms are integrated to form organized mPFC and amygdalar neuronal population activity that cooperatively controls social behavior.

Neurophysiological signatures representing neuronal population activity are local field potential (LFP) signals, consisting of diverse oscillatory patterns that dynamically vary with attentional, motivational, arousal states, and entrain synchronous rhythmic spikes (*Buzsáki, 2006*). Recently, LFP oscillations in the PFC have been shown to modulate social behavior. A PFC oscillation at a low gamma-range (20–50 Hz) band mediated by interneurons facilitates social interaction (*Liu et al., 2020*). Consistently, autism mouse models with social deficits exhibit impairments in PFC interneuronal activity (*Han et al., 2012*) and gamma oscillations (*Cao et al., 2018*). These studies suggest a key role of PFC gamma-range signals in the modulation of social behavior. On the other hand, several studies have shown that theta-range (4–12 Hz) LFP signals in the prefrontal-amygdalar circuit are associated with the expression of emotional behavior (*Calıskan and Stork, 2019*) such as fear retrieval (*Seidenbecher et al., 2003*; *Likhtik et al., 2014*; *Stujenske et al., 2014*; *Dejean et al., 2016*; *Karalis et al., 2016*; *Ozawa et al., 2020*) and anxiety (*Adhikari et al., 2010*; *Likhtik et al., 2014*). In addition, mPFC LFP power at a theta frequency (2–7 Hz) band influences oscillatory activity in the amygdala and ventral tegmental area during stress experiences (*Hultman et al., 2016*) and predicts vulnerability to mental stress in individual animals (*Kumar et al., 2014*). While these studies imply that social behavior is mediated by oscillatory signals at various frequency bands in the prefrontal-amygdalar circuit, their causal relationship and pathological changes remain fully elusive. Addressing these issues is critical for a unified understanding of neurophysiological mechanisms at a neuronal network level underlying social behavior and its deficits.

In this study, we analyzed changes in LFP signals from the dorsal mPFC (dmPFC) and basolateral amygdala (BLA) among wild-type mice and mouse models with social behavioral deficits in a social interaction (SI) test. By extracting detailed animal's behavioral patterns on a moment-to-moment basis that potentially reflect increased and decreased motivation for social behavior, we discovered prominent changes in dmPFC-BLA LFP signals that specifically varied with social behavior. Optogenetic experiments verified a causal relationship between these oscillatory signals and social behavior, highlighting the importance of frequency-specific manipulations of neuronal activity in the dmPFC-BLA circuit.

## Results

### Changes in dmPFC and BLA LFP power in a social interaction test

Male C57BL/6 J mice were tested in a conventional SI test in which they freely interacted with an empty cage and the same cage containing a target CD-1 mouse for 150 s, termed a no target and a target session, respectively (*Figure 1A*). The degree of social interactions for each mouse was quantified as a SI ratio, which refers to the ratio of stay duration in an interaction zone (IZ) in a target session to that in a no target session. Consistent with previous observations (*Golden et al., 2011*; *Venzala et al., 2012*; *Ramaker and Dulawa, 2017*), the majority of wild-type mice (11 out of 14) exhibited SI ratios of more than 1 (*Figure 1B*), demonstrating their motivation for social interactions. From the mice performing the SI tests, LFP signals were simultaneously recorded from the dmPFC, corresponding to the prelimbic (PL) region, and the BLA using an electrode assembly (*Figure 1C and D* and *Figure 1—figure supplement 1*). The locations of individual electrodes were confirmed by a postmortem histological analysis. To compute an overall tendency of LFP power changes, a fourier transformation analysis was applied to LFP signals from each entire session. Absolute LFP power spectrums were variable across individual mice (*Figure 1—figure supplement 2A*), but their averages over all mice exhibited differences between the two sessions at relatively lower (below 10 Hz) and higher (10 Hz) frequency bands (*Figure 1E*, top; n=14 mice). To further examine these differences, two LFP spectrums from the two sessions were converted to a spectrum representing the ratios of LFP power at each frequency band in the target session relative to that in the no target session (*Figure 1E*, bottom). The spectrum revealed an increase and a decrease in dmPFC power at a frequency band of 4–7 Hz and 30–60 Hz, respectively, in a target session compared with a no target session. Overall, dmPFC 4–7 Hz and 30–60 Hz power in the target session was significantly increased to 113.1 ± 4.0% and decreased to 94.6 ± 1.6%, respectively (*Figure 1F*, n=14 mice, 4–7 Hz: $t_{13}$=3.56, p=0.0070; 30–60 Hz: $t_{13}$ = 2.99, p=0.021, paired *t*-test followed by Bonferroni correction), and BLA 4–7 Hz was significantly increased

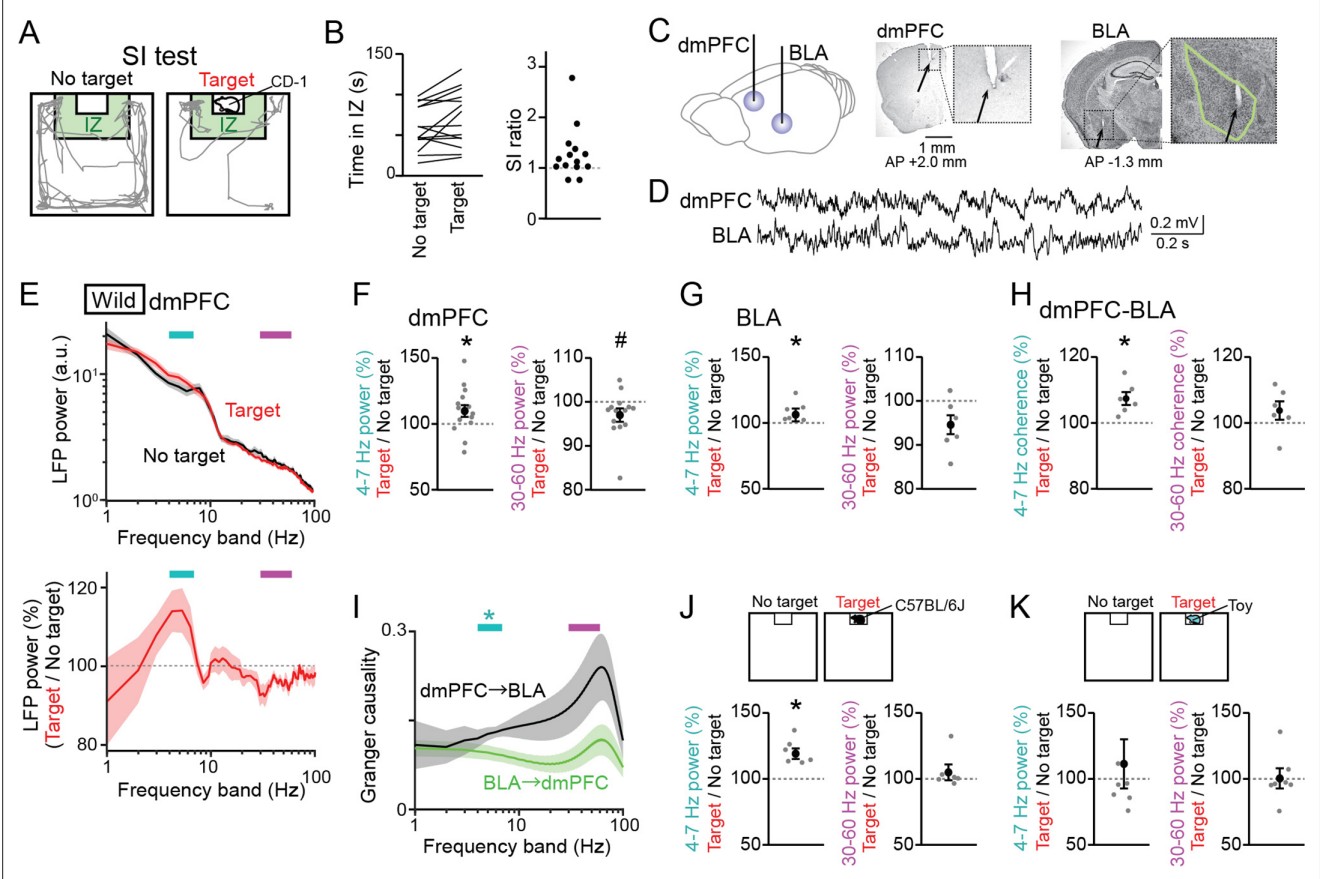

**Figure 1.** Changes in dorsal medial prefrontal cortex (dmPFC) and basolateral amygdala (BLA) Local field potential (LFP) signals in a social interaction (SI) test. (**A**) A SI test with an interaction zone (IZ; labeled in green). Movement trajectories (gray lines) of a wild-type mouse are superimposed. (**B**) (Left) Occupancy time in the IZ. Each line indicates an individual mouse (n = 14 wild-type mice). (Right) SI ratios computed from the occupancy time. Each dot represents an individual mouse. (**C**) (Left) LFPs were recorded from the dmPFC and BLA. (Right) Histological confirmation of electrode locations (arrows). The dotted boxes are magnified in the right panels. The green line shows the contour of the BLA. The details of electrode locations are shown in *Figure 1—figure supplement 1*. (**D**) Typical LFP signals from the dmPFC and BLA. (**E**) (Top) Comparison of dmPFC LFP power spectrograms between the target (red) and no target (black) sessions averaged over all mice (n = 14 mice). Original datasets from individual mice are shown in *Figure 1—figure supplement 2A*. Data are presented as the mean ± SEM. Cyan and magenta bars above represent 4–7 Hz and 30–60 Hz bands, respectively. (Bottom) The percentages of LFP power at individual frequency bands in the target session relative to those in the no target session. The percentages were computed in individual mice and were averaged over all mice. (**F**) The percentages of dmPFC 4–7 Hz (left) and 30–60 Hz (right) LFP power averaged over an entire period of the target session relative to those of the no target session (n = 14 mice). Data are presented as the mean ± SEM. Each gray dot represents an individual data points. * and # represent a significant increase and decrease in the target session, respectively (p<0.05, paired *t*-test vs no target). (**G**) Same as F but for the BLA (n = 6 mice). (**H**) Same as F but for dmPFC-BLA coherence (n = 6 mice). (**I**) Spectral granger causality averaged over dmPFC-BLA electrode pairs. (n = 6 mice). *p<0.05, Mann-Whitney *U* test followed by Bonferroni correction. (**J, K**) Same as F but when an unfamiliar C57BL/6J mouse was used as a target mouse (**J**) or a toy mouse was placed in the cage instead of a target mouse (**K**).

The online version of this article includes the following source data and figure supplement(s) for figure 1:

**Source data 1.** Individual data for *Figure 1*.

**Figure supplement 1.** Confirmation of recording sites.

**Figure supplement 2.** Datasets for local field potential (LFP) power analyses in *Figure 1E–G*.

**Figure supplement 3.** Analyses for the frequency bands other than 4–7 Hz and 30–60 Hz.

to 110.6 ± 2.7%, compared with that in the no target session (*Figure 1G*, n=6 mice, 4–7 Hz: $t_5$=4.95, p=0.0086; 30–60 Hz: $t_5$ = 2.13, p=0.17, paired *t*-test followed by Bonferroni correction). The same power analyses were applied to the same datasets at the other frequency bands, including 1–4 Hz, 7–10 Hz, 10–30 Hz, and 60–100 Hz bands, but no significant differences were found between the target and no target sessions (*Figure 1—figure supplement 3A and B*; p>0.05, paired *t*-test followed by Bonferroni correction at all the frequency bands). These results suggest that dmPFC-BLA 4–7 Hz

and 30–60 Hz power specifically become higher and lower, respectively, when mice are exposed to an environment including the other target mouse. Unlike fear-related theta-gamma coupling reported previously (*Stujenske et al., 2014*), we found no pronounced phase-amplitude coupling between the 4–7 Hz and 30–60 Hz frequency bands in both the dmPFC and BLA LFP traces (a representative result shown in *Figure 1—figure supplement 2B*). In addition to the LFP power changes, dmPFC-BLA coherence at the 4–7 Hz band in the target session was significantly higher than that in the no target session (*Figure 1H*, n=6 mice, 4–7 Hz: $t_5$=3.95, p=0.022; 30–60 Hz: $t_5$ = 1.37, p=0.46, paired *t*-test followed by Bonferroni correction), confirming the coordination of dmPFC-BLA at the 4–7 Hz band. The granger causality spectrum exhibited a significantly higher granger causality index at the 4–7 Hz band for the direction from the dmPFC to BLA than that for the direction from the BLA to the dmPFC (*Figure 1I*, n=6 mice, 4–7 Hz: p=0.030; 30–60 Hz: p=0.26, Mann-Whitney *U* test followed by Bonferroni correction), possibly reflecting the preferential projection of the dmPFC to the BLA (*Gabbott et al., 2005*; *Bukalo et al., 2015*).

As the percentages of the power changes were variable across the wild-type mice (*Figure 1F*), we examined whether the dmPFC LFP power changes in the target session in individual mice were related to their SI ratios (*Figure 1—figure supplement 2C*). However, we found no significant correlations between these two variables (n=14 mice, 4–7 Hz: $R$ = –0.27, p=0.34; 30–60 Hz: $R$=0.40, p=0.15), demonstrating that individual differences in social behavior are not crucially associated with dmPFC power changes at least within the wild-type mouse group.

A possible explanation for these power changes between the two sessions may be due to differences in running speed. To test this possibility, we compared moving speed between the two sessions and found that moving speed was significantly higher in the no target session than in the target session (*Figure 1—figure supplement 2D*; $Z$=20.20, p=9.0 × 10$^{-89}$, Mann-Whitney U test). We then compared LFP power changes between running periods with a moving speed of more than 5 cm/s and stop periods with a moving speed of less than 1 cm/s, which occupied 20.2 and 40.6% of entire recording periods, respectively (*Figure 1—figure supplement 2E and F*). Both in the dmPFC and BLA, 4–7 Hz power during stop periods was significantly higher than that during running periods (dmPFC, $Z$=4.93, p=8.26 × 10$^{-7}$; BLA, $Z$=2.42, p=0.016, Mann-Whitney *U* test), whereas 30–60 Hz power exhibited opposite changes (dmPFC, $Z$=2.13, p=0.033, p=0.19; BLA, $Z$=2.00, p=0.045), suggesting that locomotion is a crucial factor to affect these LFP power changes. We thus applied the same power analysis by specifically extracting stop and running periods. Similar to *Figure 1F*, significant increases in dmPFC LFP power in the target session were observed during stop periods but not running periods (*Figure 1—figure supplement 2H*, stop periods: n=14 mice, 4–7 Hz: $t_{13}$=3.72, p=0.0052; 30–60 Hz: $t_{13}$ = 3.05, p=0.019; *Figure 1—figure supplement 2G*, running periods: 4–7 Hz: $t_{13}$=1.33, p=0.41; 30–60 Hz: $t_{13}$ = 1.25, p=0.46, paired *t*-test followed by Bonferroni correction). These results confirm that while 4–7 Hz and 30–60 Hz power in the dmPFC and BLA is higher and lower, respectively, as moving speed is lower, the LFP power changes were still prominent in the entire target session, compared with the no target session, when mice stopped.

In the SI test above, we utilized a CD-1 mouse as a target mouse in the cage that was substantially larger than the recorded C57BL/6 J mice. This recording condition may induce anxiety-related or fear-related behavior in the recorded mice. To reduce these emotional factors as possible, we performed a similar SI test using an unfamiliar C57BL/6 J mouse with a similar body size as a target mouse (*Figure 1J*). Similar to the results when the target CD-1 mice were used (*Figure 1F*), dmPFC 4–7 Hz power in the target session was significantly higher than that in the no target session (*Figure 1J*, n=6 mice, 4–7 Hz: $t_5$=5.36, p=0.0060; 30–60 Hz: $t_5$ = 1.04, p=0.70, paired *t*-test followed by Bonferroni correction). In addition, as a control experiment without social behavior, we performed a similar SI test by placing a plastic toy mouse in the cage as a novel object instead of a real mouse (*Figure 1K*). In this case, significant changes in dmPFC 4–7 Hz and 30–60 Hz power between the two sessions were not observed (*Figure 1K*, n=7 mice, 4–7 Hz: $t_6$=0.51, p>0.99; 30–60 Hz: $t_6$ = 0.60, p>0.99, paired *t*-test followed by Bonferroni correction). These results further confirm that the dmPFC power changes are induced specifically in a condition where mice exhibit social behavior while they are less associated with an object novelty or emotional components such as anxiety.

While LFP power changes were observed in the target session with both a CD-1 mouse and a C57BL/6 J mouse as a target mouse (*Figure 1F and J*), there still remains a possibility that these changes are induced by increased anxiety and/or novelty against social interaction, as mice are

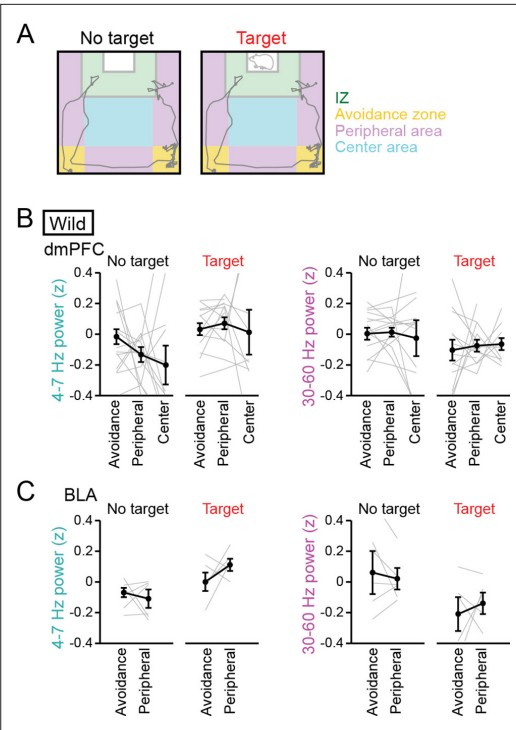

**Figure 2.** No pronounced changes in 4–7 Hz and 30–60 Hz power in the dorsal medial prefrontal cortex (dmPFC) and basolateral amygdala (BLA) in areas outside the interaction zone (IZ). (**A**) Schematic illustration showing social avoidance zones, peripheral areas, and a center area in the SI test. (**B**) Comparisons of dmPFC 4–7 Hz and 30–60 Hz power across avoidance zones, peripheral areas, and a center area (n = 14 mice). Data are presented as the mean ± SEM. Each line represents each mouse. p>0.05, Mann-Whitney *U* test followed by Bonferroni correction. (**C**) Same as B but for the BLA (n = 6 mice). The center area was removed from this analysis because of the limited number of samples.

The online version of this article includes the following source data and figure supplement(s) for figure 2:

**Source data 1.** Individual data for *Figure 2*.

**Figure supplement 1.** Supplementary datasets for analyses of socially deficient mice.

generally anxious when encountering a novel (target) mouse. Previous studies have demonstrated that anxiety induces theta-range (4–10 Hz) power increases in the mPFC-BLA-ventral hippocampal circuit (*Adhikari et al., 2010*; *Likhtik et al., 2014*; *Padilla-Coreano et al., 2019*). We thus examined whether anxiogenic conditions could induce the similar changes in the dmPFC-BLA LFP signals observed in this study. The test box was divided into social avoidance zones (corners of the box opposing the target mice), peripheral areas (near the walls of the box), and a center area (*Figure 2A*). Based on the similarity of the no target session and conventional open field tests, mice are considered to more feel anxiety in the avoidance zones and peripheral areas, compared with the center area, in the no target session. To compare relative changes in LFP power across behavior and sessions, LFP power at each frequency band was z-scored based on the average and SD of LFP power at each frequency band in an entire period including the no target and target sessions. In the no target session, no significant differences in dmPFC and BLA 4–7 Hz and 30–60 Hz LFP power were observed among these areas (*Figure 2B*, dmPFC: n=7 mice, 4–7 Hz, $F_{2,40}$ = 1.36, p=0.27; 30–60 Hz; $F_{2,40}$ = 0.11, p=0.90; *Figure 2C*, BLA: n=6 mice, 4–7 Hz, $F_{2,16}$ = 1.12, p=0.35; 30–60 Hz; $F_{2,16}$ = 0.61, p=0.56, one-way ANOVA). On the other hand, mice are considered to most increase anxiety or most decrease motivation for social behavior in the avoidance zones (*Golden et al., 2011*) and more increase anxiety levels in the peripheral areas, compared with the center area, in the target session. In the target session, no significant differences in dmPFC and BLA 4–7 Hz and 30–60 Hz LFP power were observed among these areas (*Figure 2B*, dmPFC: n=14 mice, 4–7 Hz, $F_{2,35}$ = 0.50, p=0.61; 30–60 Hz; $F_{2,35}$ = 0.13, p=0.88; *Figure 2C*, BLA: n=6 mice, 4–7 Hz, $F_{1,10}$ = 3.38, p=0.099; 30–60 Hz; $F_{1,10}$ = 0.27, p=0.62, one-way ANOVA). These results suggest that anxiety-related environments are not crucially associated with 4–7 Hz and 30–60 Hz LFP power changes in the dmPFC and BLA observed in this study.

## Increases in dmPFC 4–7 Hz power during social avoidance in socially deficient mouse models

We next examined whether these LFP signals are altered in mice with reduced social interaction. The same tests were performed on Shank3 null mutant mice, termed Shank3 knockout (KO) mice, which have been reported to exhibit repetitive grooming behavior and social interaction deficits, mimicking symptoms associated with ASD (*Peça et al., 2011*; *Mei et al., 2016*). The SI ratios in 7 Shank3 KO mice were significantly lower than those in the 14 wild-type mice (*Figure 3A*, Z=2.65, p=0.016, Mann-Whitney *U* test followed by Bonferroni correction). We recorded LFP signals from the dmPFC and

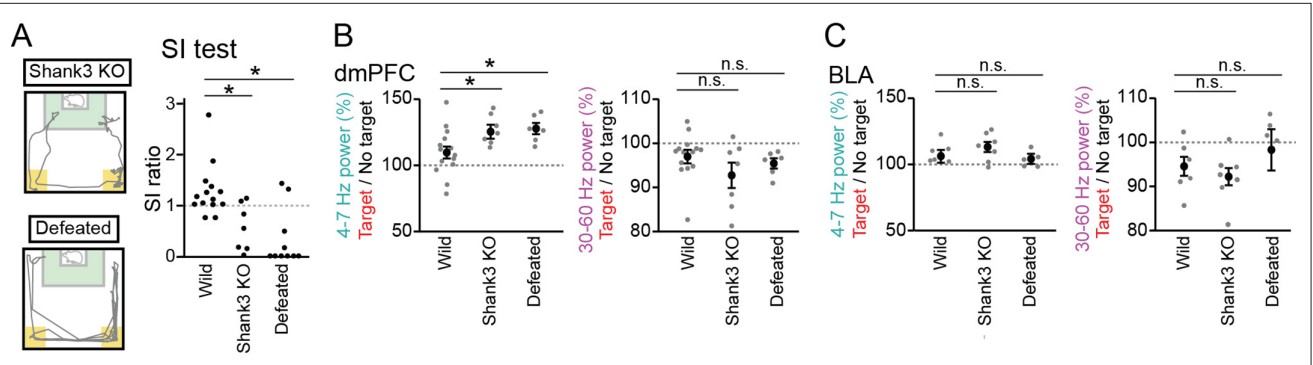

**Figure 3.** Further increases in dorsal medial prefrontal cortex (dmPFC) 4–7 Hz power in the target session in Shank3 knockout (KO) mice and defeated mice. (**A**) (Left) Movement trajectory of a Shank3 KO mouse and a defeated mouse in a target session. The orange areas represent the avoidance zones. (Right) SI ratios for Shank3 KO and defeated mice (n = 14 wild, 7 Shank3 KO, and 10 defeated mice). Each dot represents an individual animal. The data from wild-type mice similar to those shown in *Figure 1B* are presented for comparison. *p<0.05 versus wild, Mann-Whitney *U* test followed by Bonferroni correction. (**B**) The percentages of dmPFC 4–7 Hz (left) and 30–60 Hz (right) local field potential (LFP) power in the target session relative to those in the no target session (n = 14 wild, 7 Shank3 KO, and 6 defeated mice). Data are presented as the mean ± SEM. Each gray dot represents an individual data points. The data from wild-type mice similar to those shown in *Figure 1F* are presented for comparison. *p<0.05, versus wild, Mann-Whitney *U* test followed by Bonferroni correction. (**C**) Same as B but for the basolateral amygdala (BLA) (n = 6, 7, and 5 mice).

The online version of this article includes the following source data for figure 3:

**Source data 1.** Individual data for *Figure 3*.

BLA of these Shank3 KO mice and found significant increases in dmPFC and BLA LFP power during the target session at the 4–7 Hz bands, similar to the wild-type mice (*Figure 3B*, dmPFC: n=7 mice, 4–7 Hz, $t_6$=6.04, p=1.8 × 10$^{-3}$; 30–60 Hz; $t_6$=2.62, p=0.078; *Figure 3C*, BLA: n=7 mice, 4–7 Hz, $t_6$=3.03, p=0.048; 30–60 Hz; $t_6$=3.52, p=0.026, paired *t*-test followed by Bonferroni correction). Moreover, the dmPFC 4–7 Hz increases during the target session in the Shank3 KO mice were significantly larger than those observed in the wild-type mice (*Figure 3B*, $F_{2,24}$=4.21, p=0.027, one-way ANOVA across wild-type, Shank3 KO, and defeated mouse groups; Z=2.36, p=0.046, Mann-Whitney *U* test followed by Bonferroni correction), whereas no differences were observed for the changes in dmPFC 30–60 Hz power ($F_{2,24}$=1.55, p=0.23, one-way ANOVA; Z=1.23, p=0.44, Mann-Whitney *U* test followed by Bonferroni correction) and BLA power (*Figure 3C*, 4–7 Hz, $F_{2,15}$=1.07, p=0.36, one-way ANOVA; p=0.90; 30–60 Hz, $F_{2,15}$=1.01, p=0.39, one-way ANOVA; P>0.99, Mann-Whitney *U* test followed by Bonferroni correction). These results suggest that the increases in dmPFC 4–7 Hz power during a target session are more prominent in Shank3 KO mice, compared with wild-type mice.

During the target session, the Shank3 KO mice spent substantial (26.0 ± 6.8%) time in the avoidance zones (*Figure 3A* and *Figure 2—figure supplement 1A*). When the Shank3 KO mice stayed within the avoidance zone, dmPFC 4–7 Hz was significantly increased, compared with the other areas (*Figure 4B*, dmPFC: n=7 mice, 4–7 Hz, $t_5$=4.39, p=0.014; 30–60 Hz; $t_5$=0.33, p>0.99; BLA: n=7 mice, 4–7 Hz, $t_5$=1.62, p=0.32; 30–60 Hz; $t_5$=0.17, p>0.99, paired *t*-test followed by Bonferroni correction). Such significant changes were not observed in the wild-type mice (*Figure 4A*, dmPFC: n=13 mice that stayed in the avoidance zones, 4–7 Hz, $t_{12}$=0.068, p>0.99; 30–60 Hz; $t_{12}$=0.71, p=0.98; BLA: n=5 mice that stayed in the avoidance zones, 4–7 Hz, $t_4$=0.49, p>0.99; 30–60 Hz; $t_4$=0.27, p>0.99, paired *t*-test followed by Bonferroni correction). These results demonstrate that Shank3 KO mice specifically exhibit a dmPFC 4–7 Hz power increase during avoidance behavior. In the Shank3 KO mice, dmPFC-BLA coherence and the directionality between the dmPFC and BLA at the 4–7 Hz band were not prominent during social avoidance behavior (*Figure 2—figure supplement 1B*, n=6 mice, 4–7 Hz: $t_5$=0.46, p>0.99; 30–60 Hz: $t_5$ = 0.98, p=0.73, paired *t*-test followed by Bonferroni correction; *Figure 2—figure supplement 1C*, n=6 mice, 4–7 Hz: p=0.63; 30–60 Hz: p>0.99, Mann-Whitney *U* test followed by Bonferroni correction).

We next tested whether socially defeated mice with reduced social interaction exhibit similar LFP changes. Wild-type mice were exposed to social defeat stress for 10 consecutive days, termed defeated mice. SI ratios of the 6 defeated mice were significantly lower than those in the 14 wild-type mice (*Figure 3A*; Z=4.10, p=8.1 × 10$^{-5}$, Mann-Whitney *U* test followed by Bonferroni correction).

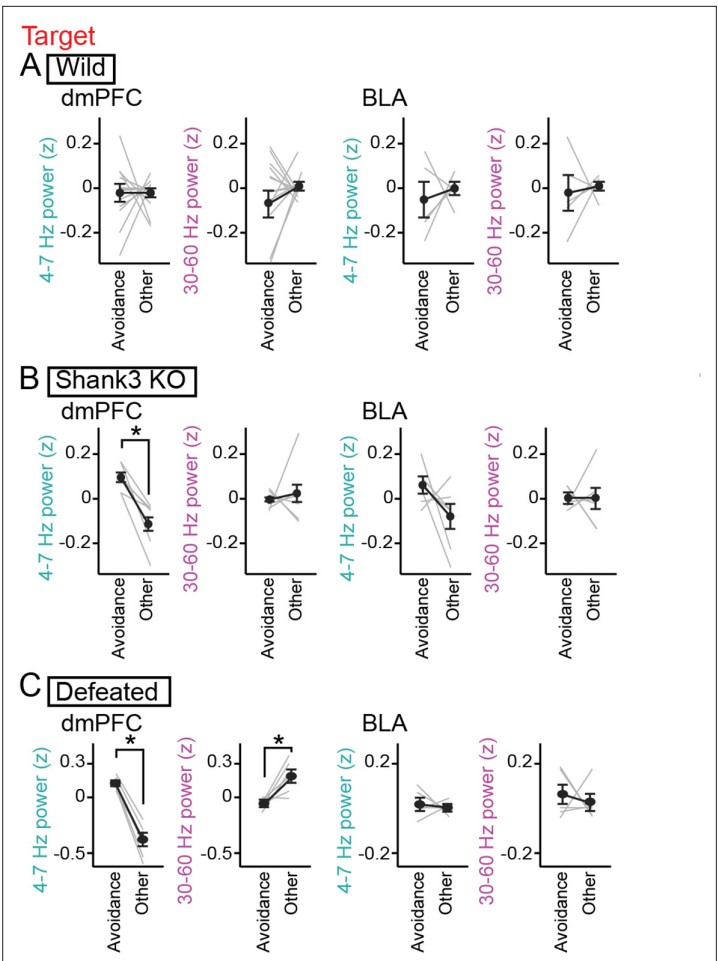

**Figure 4.** Increases in dorsal medial prefrontal cortex (dmPFC) 4–7 Hz power during social avoidance in Shank3 knockout (KO) mice and defeated mice. (**A**) Comparisons of 4–7 Hz and 30–60 Hz power in the dmPFC and basolateral amygdala (BLA) between the avoidance zones and the other areas in the target session in wild-type mice (n = 14 and 6 mice). Data are presented as the mean ± SEM. Each gray line represents each mouse. p>0.05, paired *t*-test followed by Bonferroni correction. (**B**) Same as A but for Shank3 KO mice (n = 7 and 7 mice). *p<0.05, paired *t*-test followed by Bonferroni correction. (**C**) Same as A but for defeated mice (n = 6 and 5 mice).

The online version of this article includes the following source data for figure 4:

**Source data 1.** Individual data for **Figure 4**.

Similar to the wild-type and Shank3 KO mice, these defeated mice exhibited a significantly larger increase in dmPFC 4–7 Hz power during the target session than during the no target session (**Figure 3B**, dmPFC: n=6 mice, 4–7 Hz, $t_5$=6.93, p=9.6 × 10$^{-4}$; 30–60 Hz; $t_5$=3.77, p=0.013; BLA: n=5 mice, 4–7 Hz: $t_4$=1.88, p=0.14; 30–60 Hz: $t_4$=0.33, p=0.76). In addition, similar to the Shank3 KO mice, the dmPFC 4–7 Hz increase in the defeated mice was significantly larger than that observed from the wild-type mice (**Figure 3B**, dmPFC: 4–7 Hz: Z=2.27, p=0.046; 30–60 Hz: Z=1.36, p=0.34; **Figure 3C**, BLA: 4–7 Hz: p>0.99; 30–60 Hz: p=0.50, Mann-Whitney *U* test followed by Bonferroni correction). Similar significant results were observed in the comparison between the defeated mice and defeated control mice (that were pair housed in the same cage with aggressor mice but not subject to physical contact) (**Figure 2—figure supplement 1E** and **Figure 4F**, 4–7 Hz: p=0.0087; 30–60 Hz: p=0.13, Mann-Whitney *U* test followed by Bonferroni correction). Furthermore, defeated mice spent 54.7 ± 9.5% of an entire recording time in the avoidance zones (**Figure 3A** and **Figure 2— figure supplement 1A**) and exhibited significant increases in dmPFC 4–7 Hz power in the avoidance zone (**Figure 4C**, dmPFC: n=6 mice, 4–7 Hz, $t_5$=11.68, p=1.8 × 10$^{-4}$; 30–60 Hz; $t_5$=3.45, p=0.036; BLA: n=5 mice, 4–7 Hz, $t_4$=0.47, p>0.99; 30–60 Hz; $t_4$=0.46, p>0.99, paired *t*-test followed by Bonferroni

correction). These results demonstrate that a dmPFC 4–7 Hz power increase during social avoidance also occurs in depression model mice, similar to Shank3 KO mice. Taken together, our results from the two mouse models suggest that dmPFC 4–7 Hz power increases during social avoidance behavior are a common hallmark across socially deficient mouse models.

## Changes in dmPFC LFP power during social approach behavior

The social avoidance-related increases in dmPFC 4–7 Hz power implied that social interaction behavior may be associated with dmPFC 4–7 Hz power changes. To test this possibility, we first compared LFP power between when the wild-type normal mice stayed within and were outside the IZ as conventional measures in an SI test. However, no significant changes in 4–7 Hz and 30–60 Hz power in the dmPFC and BLA during the target session were detected between the IZ and the other areas (*Figure 5—figure supplement 1*; dmPFC, n=14 mice; 4–7 Hz: $t_{13} = 0.76$, p=0.46; 30–60 Hz: $t_{13} = 1.38$, p=0.19; BLA, n=6 mice; 4–7 Hz: $t_5 = 0.76$, p=0.48; 30–60 Hz: $t_5 = 0.02$, p=0.98, paired *t*-test followed by Bonferroni correction). While this analysis focused on the entire period during which the mice stayed in the IZ, their behavioral patterns within the IZ were not consistent across time; mice actively approach or interact with a target mouse in some periods, reflecting high motivation, whereas they occasionally turn around, move away from a target mouse, or continue to stay at a location in an IZ, possibly reflecting no strong motivation for social interaction. These behavioral observations indicate that animals' motivation toward and salience regarding the other mouse are not equivalent even when they are similarly located in an IZ.

The results suggest that changes in dmPFC-BLA oscillations are not simply explained by where the mice stayed in the SI test. We further analyzed how LFP patterns are associated with their

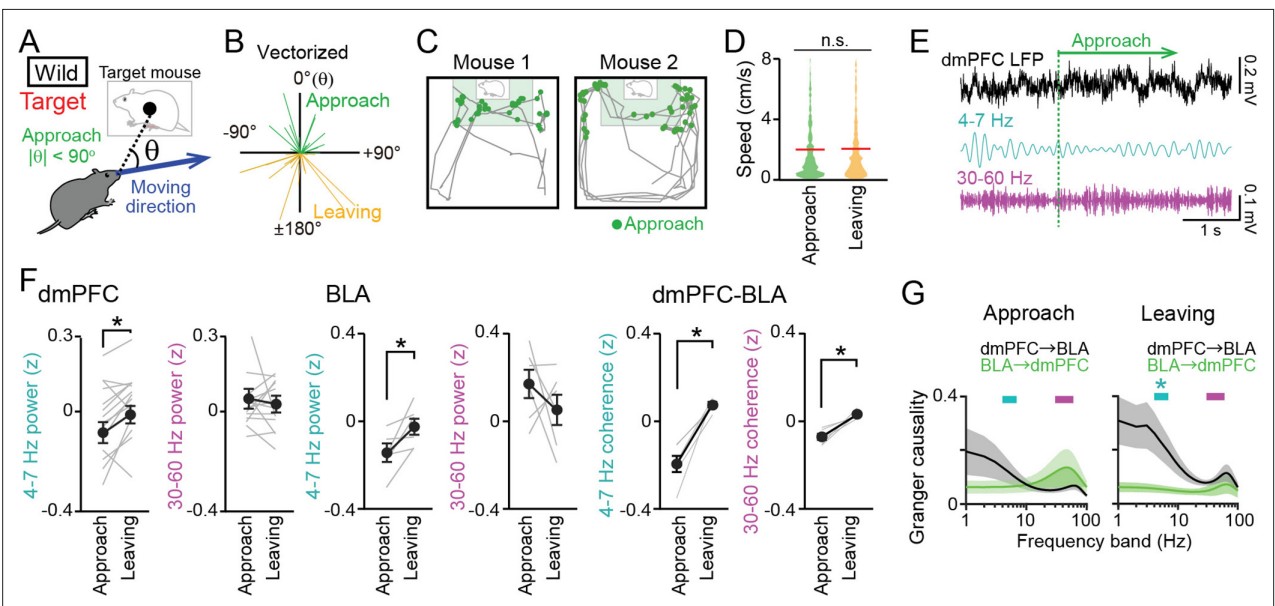

**Figure 5.** Decreases in dorsal medial prefrontal cortex (dmPFC) 4–7 Hz local field potential (LFP) power during social approach behavior in the target session. (**A**) Social approach and leaving behavior in wild-type mice was defined when absolute cage-oriented moving directions (| θ |) were less than and more than 90°, respectively, in the half of the box containing the interaction zone (IZ). (**B**) A polar plot of vectorized instantaneous animal trajectories (bin = 1 s) as a function of cage-oriented moving direction. (**C**) Trajectories from two representative mice (gray). Green dots represent social approach behavior. (**D**) Distributions of moving speed during approach and leaving behavior (n = 644 and 577). The red lines show the average. p>0.05, Mann-Whitney *U* test. (**E**) Unfiltered and bandpass (4–7 Hz and 30–60 Hz)-filtered dmPFC LFP traces in the target session. The green line indicates the onset of a social approach behavior. (**F**) Comparison of 4–7 Hz and 30–60 Hz power in the dmPFC (left, n = 14 mice) and basolateral amygdala (BLA) (middle, n = 6 mice) and dmPFC-BLA 4–7 Hz and 30–60 Hz coherence (right, n = 6 mice) between approach and leaving behavior. Data are presented as the mean ± SEM. Each gray line represents each mouse. *p<0.05, paired *t*-test. (**G**) Spectral Granger causality during approach (top) and leaving (bottom) behavior in the target session averaged over dmPFC-BLA electrode pairs. *p<0.05, Mann-Whitney *U* test followed by Bonferroni correction.

The online version of this article includes the following source data and figure supplement(s) for figure 5:

**Source data 1.** Individual data for *Figure 5*.

**Figure supplement 1.** local field potential (LFP) power changes in the interaction zone in the target session.

instantaneous behavior every 1 s. We defined social approach behavior, potentially representing increased motivation for social interaction, as the time during which the mice approached the cage (within the half of the box containing the cage) with their cage-oriented moving directions θ less than 90° in the target session (*Figure 5A–C*). Assuming that mice potentially exhibited the highest motivation and salience during an initial bout of a social approach, this definition was restricted to the initial 5 s periods of social approach behavior. As a control behavior against approach behavior, we defined leaving behavior as the time during which the mice left from the cage with their cage-oriented moving directions θ more than 90° in the target session. No significant differences in the distributions of moving speed were found between the approach behavior and leaving behavior (*Figure 5D*; $Z$=0.85, p=0.39, Mann-Whitney $U$ test). These results confirm that approach and leaving behavior is not explained by speed or locomotion itself, allowing us to compare LFP power between the two behavioral periods without being affected by moving speed. Both in the dmPFC and BLA, 4–7 Hz power was significantly decreased during the approach behavior compared with the leaving behavior (*Figure 5F*, left, dmPFC: 4–7 Hz: $t_{13}$=2.71, p=0.018; 30–60 Hz: $t_{13}$=0.47, p=0.65; *Figure 5F*, middle, BLA: 4–7 Hz: $t_5$=2.83, p=0.037; 30–60 Hz: $t_5$=1.01, p=0.36, paired $t$-test). No significant differences in LFP power at the other frequency bands (1–4 Hz, 7–10 Hz, 10–30 Hz, and 60–100 Hz bands) were observed between the approach and leaving behavior (*Figure 1—figure supplement 3C and D*, p>0.05, paired $t$-test followed by Bonferroni correction, target vs no target). Consistent with the LFP power changes, dmPFC-BLA coherence at the 4–7 Hz band during leaving behavior was significantly higher than that during approach behavior (*Figure 5F*, right, n=6 mice, 4–7 Hz: $t_5$=6.34, p=0.0028; 30–60 Hz: $t_5$ = 5.54, p=0.0052, paired $t$-test followed by Bonferroni correction). Moreover, the Granger causality index at the 4–7 Hz band in the dmPFC-BLA direction was significantly higher than that in the BLA-dmPFC direction during leaving behavior, but not during approach behavior (*Figure 5G*, n=6 mice, 4–7 Hz: p=0.030; 30–60 Hz: p=0.26, Mann-Whitney $U$ test followed by Bonferroni correction). These results suggest that functional information transfer at the 4–7 Hz band from the dmPFC to the BLA is lowered during social approach behavior, compared with leaving behavior. In Shank3 KO mice, we did not observe significant differences in dmPFC 4–7 Hz and 30–60 Hz power between approach and leaving behavior (*Figure 2—figure supplement 1D*, n=7 mice, 4–7 Hz: $t_6$=0.48, p=0.65; 30–60 Hz: $t_6$=0.67, p=0.53, paired $t$-test followed by Bonferroni

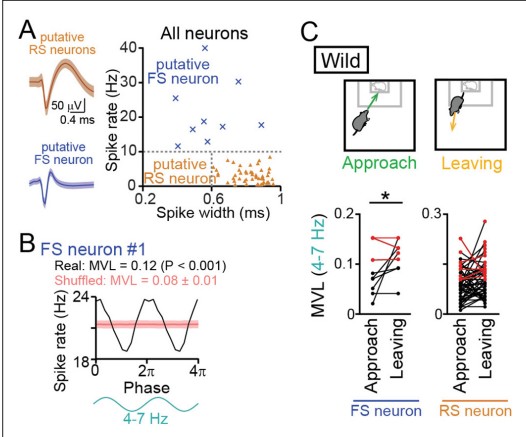

**Figure 6.** Entrainment of oscillatory spike patterns of dorsal medial prefrontal cortex (mPFC) neurons. (**A**) (Left) Typical spike waveforms of a putative regular-spiking (RS) neuron (top) and a putative fast-spiking (FS) neuron (bottom). Data are presented as the mean ± SD. (Right) For individual neurons in wild-type mice, baseline spike rates and spike width are plotted. Each dot represents an individual neuron. Neurons plotted in orange and cyan regions are classified as putative excitatory RS pyramidal neurons (triangle, $n$ = 48 neurons) and inhibitory FS interneurons (cross-mark, $n$ = 9 neurons), respectively. (**B**) A representative putative FS neuron that fired time-locked to the 4–7-Hz local field potential (LFP) oscillations. Instantaneous spike rates of this neuron are plotted against the phase of 4–7-Hz oscillatory cycles. From a phase-spike distribution, a mean vector length (MVL) was computed as 0.12. The red line and the shaded area represent the mean and SD computed from the corresponding 1000 shuffled datasets. (**C**) Comparisons of MVL for the 4–7-Hz LFP oscillations between approach and leaving behavior. RS and FS neurons were separately analyzed. Each dot and line represents each neuron. The red dots indicate significant MVL, computed from shuffled datasets. The red lines indicate neurons showing significant MVL in both of the periods, which were considered as behavior-irrelevant phase-locked neurons and excluded from the statistical analyses. p<0.05, paired $t$-test for the datasets shown in black.

The online version of this article includes the following source data and figure supplement(s) for figure 6:

**Source data 1.** Individual data for *Figure 6*.

**Figure supplement 1.** Comparisons of mean vector length (MVL) for the oscillations (at 1–4 Hz, 4–7 Hz, 7–15 Hz, 15–30 Hz, and 30–60 Hz) other than the 4–7-Hz local field potential (LFP) oscillations between approach and leaving behavior.

correction), suggesting that social behavior-related dmPFC activity is not properly regulated in socially deficient mice. The reductions of dmPFC 4–7 Hz power during social interaction behavior are consistent with the opposite changes (the increases in dmPFC 4–7 Hz power) observed during social avoidance behavior in the socially deficient mice (shown in *Figures 3 and 4*). Taken together, our results suggest that changes in dmPFC 4–7 Hz oscillations are a key neuronal substrate to modulate social behavior and social avoidance.

## Neuronal spikes associated with LFP oscillations in the mPFC

We next analyzed how these LFP oscillations entrain spikes of individual mPFC neurons, defined by single-unit isolation. Following the criteria utilized in previous studies (*Wilson et al., 1994*; *Tierney et al., 2004*; *Homayoun and Moghaddam, 2007*), as shown in *Figure 6A*, putative regular-spiking (RS) excitatory pyramidal neurons were identified as neurons that had baseline spike rates lower than 10 Hz and spike widths longer than 0.6ms (n=48 neurons from 11 mice). On the other hand, putative fast-spiking (FS) interneurons were identified as neurons that had baseline spike rates higher than 10 Hz (n=9 neurons from 6 mice). While these criteria might define a minority of interneurons as putative RS neurons, it was unlikely that true pyramidal neurons are misclassified as putative FS neurons (*Wilson et al., 1994*; *Tierney et al., 2004*; *Homayoun and Moghaddam, 2007*). We tested whether these mPFC neurons exhibited spike patterns phase-locked to the 4–7 Hz oscillations (*Figure 6B*). All spike analyses were restricted to a target session. An example putative FS neuron shown in *Figure 6B* exhibited apparent spike rate changes corresponding to altering phases in the 4–7 Hz oscillations. For each neuron, the degree of spike phase locking was quantified by computing the mean vector length (MVL). In the example neuron, the MVL was 0.12. To assess the significance of each MVL, we created shuffled datasets in which spike timing was randomized within the session and MVL was similarly computed from 1000 shuffled datasets, termed $MVL_{shuffled}$. The MVL of an original data was considered to be significant (p<0.05) when the MVL was higher than the top 95% of the corresponding $MVL_{shuffled}$. According to this criterion, the MVL of the example neuron was higher than the corresponding 1000 $MVL_{shuffled}$ (p<0.001), demonstrating that these neuronal spikes were entrained by the 4–7 Hz oscillations. For each neuron, the MVL was compared between approach and leaving behavior (*Figure 6C*). Of the 48 and 9 putative RS and FS neurons tested, 5 (10.4%) and 3 (33.3%) neurons showed significant MVL in both approach and leaving behavior (the neurons indicated by the red lines in *Figure 6C*). These neurons were considered to show phase locking spikes irrespective of behavior and excluded from further statistical analyses. After this exclusion, the remaining FS neurons showed significantly higher MVL for the 4–7 Hz oscillations during leaving periods, compared with approach periods (n=6 neurons; $t_5$=3.15, p=0.025, paired *t*-test). These results demonstrate that a subset of FS neurons alter their entrainment to the 4–7 Hz oscillations depending on animal's behavioral patterns. The remaining RS neurons did not show such significant changes depending on behavioral patterns (n=43 neurons; $t_{42}$=1.69, p=0.69). The same analyses were applied to the other frequency bands (1–4 Hz, 4–7 Hz, 7–15 Hz, 15–30 Hz, and 30–60 Hz) but no significant differences were observed between approach and leaving behavioral periods (*Figure 6—figure supplement 1*; p>0.05, paired *t*-test). Taken together, these results suggest that mPFC FS neurons are more preferentially entrained by the 4–7 Hz oscillations during leaving behavior than approach behavior. In other words, the entrainment of mPFC neurons to the 4–7 Hz oscillations is disrupted when mice engage in social approach behavior.

## Restoration of social interaction by optogenetic manipulation of dmPFC 4–7 Hz power

The observations that dmPFC 4–7 Hz power was reduced during social approach behavior imply that replicating such dmPFC LFP patterns may potentially facilitate social interaction behavior. To address this idea, a technique to reduce dmPFC 4–7 Hz oscillations is needed. Based on our observations that the entrainment of mPFC inhibitory neuronal spikes to the 4–7 Hz oscillations dynamically varies with social behavior, we sought to develop a method to alter dmPFC 4–7 Hz oscillations by manipulating dmPFC inhibitory neurons. Here, we focused on parvalbumin (PV)-positive interneurons, a major type of interneurons as this cell type has been reported to be crucial for the generation of cortical gamma-range (30–60 Hz) oscillations (*Whittington et al., 1995*; *Bartos et al., 2007*; *Cardin et al., 2009*; *Sohal et al., 2009*; *Buzsáki and Wang, 2012*; *Nakamura et al., 2015*; *Cao et al., 2018*; *Liu et al., 2020*). To selectively control the activity of PV-positive interneurons, we utilized

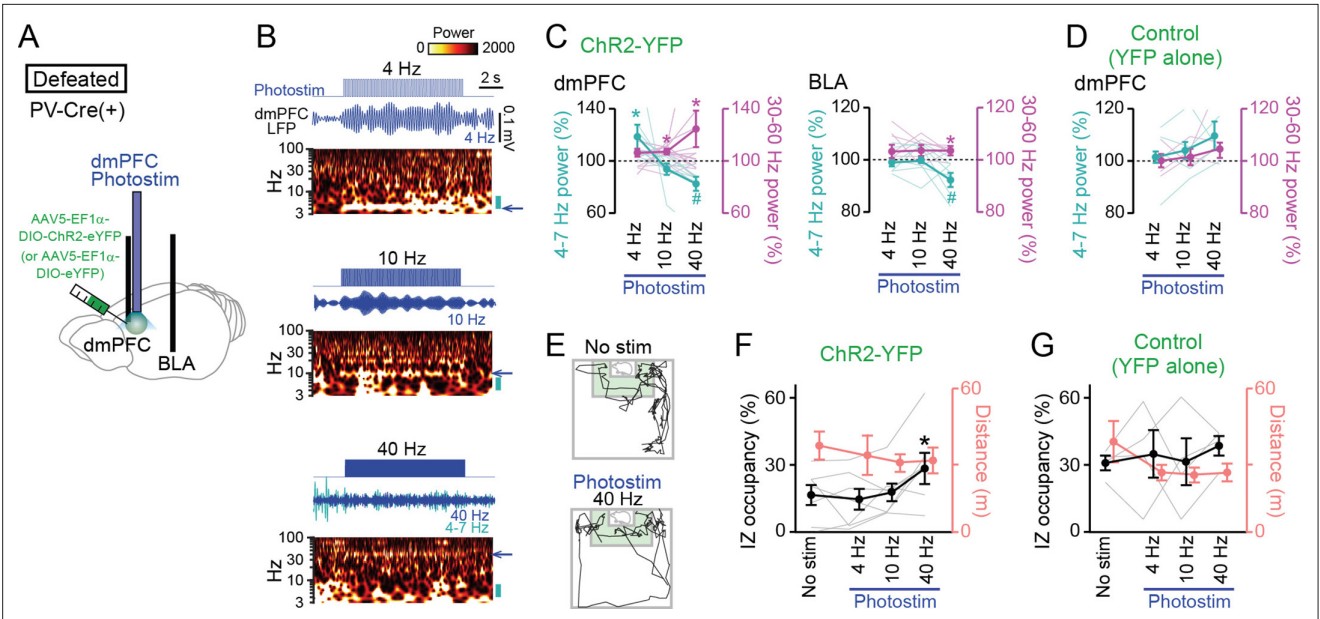

**Figure 7.** Restoration of social approach behavior by optogenetic photostimulation with a decrease in 4–7 Hz power and an increase in 30–60 Hz power in the dorsal medial prefrontal cortex (dmPFC). (**A**) Schematic illustration. PV-positive interneurons expressing ChR2-YFP (or YFP alone) in the dmPFC were optogenetically stimulated, while local field potential (LFP) signals were recorded from the dmPFC and basolateral amygdala (BLA) in PV-Cre mice subjected to social defeat stress. (**B**) (From top to bottom) Photostimulation at 4, 10, and 40 Hz for 10 s in defeated mice injected with AAV5-EF1a-DIO-ChR2-eYFP. Each panel shows photostimulation patterns (upper), representative dmPFC LFP traces filtered at the frequency band similar to that of photostimulation (middle) and its wavelet spectrum (lower). At 40-Hz photostimulation, the LFP trace filtered at 4–7 Hz is superimposed as the cyan trace. The blue arrows beside the wavelet spectrum show the corresponding band and the cyan bars represents the 4–7 Hz band. (**C**) Average 4–7 Hz (cyan) and 30–60 Hz (magenta) power changes by dmPFC photostimulation in the dmPFC (left) and BLA (right) in defeated mice injected with AAV5-EF1a-DIO-ChR2-eYFP. (n = 7 mice). Each thin line represents each mouse. * and # represent a significant increase and decrease, respectively (p<0.05, paired *t*-test versus baseline). (**D**) Same as C but for control mice injected with AAV5-EF1a-DIO-eYFP (n = 4 mice). (**E**) Movement trajectories of a defeated mouse in a target session with no stimulation and 40-Hz photostimulation. (**F**) The percentage of time spent in the IZ (left axis, black) and total travel distance (right axis, thin red) in defeated mice injected with AAV5-EF1a-DIO-ChR2-eYFP (n = 7 mice). Data are presented as the mean ± SEM. Each gray line represents each mouse. *p<0.05, paired *t*-test followed by Bonferroni correction. (**G**) Same as F but for defeated mice injected with AAV5-EF1a-DIO-eYFP (n = 5 mice).

The online version of this article includes the following source data for figure 7:

**Source data 1.** Individual data for *Figure 7*.

optogenetic tools with PV-Cre mice (*Cao et al., 2018*; *Liu et al., 2020*). While we noted that not all interneurons in the dmPFC express PV, we took this optogenetic approach as a means to potentially alter dmPFC neuronal oscillations at these frequency bands. The PV-Cre mice were first subjected to chronic social defeat stress and then injected with a Cre-inducible viral construct, AAV5-EF1a-DIO-ChR2-eYFP or AAV5-EF1a-DIO-eYFP, into the dmPFC so that PV interneurons selectively expressed ChR2-YFP or YFP, respectively (*Figure 7A*). In addition, an optic cannula and recording electrodes were implanted into the identical region of the dmPFC, and additional recording electrodes were implanted into the BLA. We first sought to identify appropriate photostimulation protocols by simultaneous LFP recordings of the dmPFC and BLA. Two weeks after surgery, photostimulation at 4, 10, and 40 Hz was applied in defeated mice expressing ChR2-YFP (*Figure 7B*), a frequency corresponding to 4–7 Hz, an intermediate frequency, and a frequency corresponding to 30–60 Hz, respectively. The width of each photostimulation pulse was set to half the pulse intervals: 125, 50, and 12.5ms for 4, 10, and 40 Hz, respectively, meaning that the total duration of applied photostimulation was equivalent in all protocols. Photostimulation at 4 Hz induced a significant 19.5 ± 9.0% power increase in the corresponding frequency (4–7 Hz) band in the dmPFC (*Figure 7B and C*; 4–7 Hz: $t_{15}$=2.44, p=0.045; 30–60 Hz: $t_{15}$=2.18, p=0.066, paired *t*-test), consistent with the entrainment of inhibitory neuronal spikes to 4–7 Hz oscillations. Photostimulation at 10 Hz induced a small (7.2 ± 2.2%) but significant power increase at the 30–60 Hz band in the dmPFC (4–7 Hz: $t_{15}$=1.40, p=0.20; 30–60 Hz: $t_{15}$=3.28, p=0.014). Photostimulation at 40 Hz induced significant 24.8 ± 14.1% and 2.8 ± 0.9% power increases

at the corresponding frequency (30–60 Hz) band in the dmPFC and BLA, respectively (*Figure 7B and C*, dmPFC: $t_{15}$=2.91, p=0.023; BLA: $t_{15}$=3.27, p=0.017). In addition, the 40 Hz photostimulation induced significant 17.5 ± 5.4% and 6.4 ± 2.6% power decreases in the 4–7 Hz band in the dmPFC and BLA, respectively (*Figure 7C*, dmPFC: $t_{15}$=3.24, p=0.014; BLA: $t_{15}$=2.49, p=0.047). The bidirectional changes in 4–7 Hz and 30–60 Hz power in the dmPFC-BLA circuit at 40 Hz photostimulation are consistent with those observed during social approach behavior, as shown in *Figure 5*, suggesting that it was possible to replicate social interaction-related neuronal activity. Mice expressing YFP alone showed no significant changes in dmPFC 4–7 Hz and 30–60 Hz power by any photostimulation conditions (*Figure 7D*; n=4 mice; 4 Hz: $t_3$=0.043, p=0.97; 10 Hz: $t_3$=1.58, p=0.21; 40 Hz: $t_3$=1.56, p=0.22; 4 Hz: $t_3$=0.35, p=0.74; 10 Hz: $t_3$=0.74, p=0.52; 40 Hz: $t_3$=1.88, p=0.16, paired *t*-test).

The defeated PV-Cre mice were tested in a target session with these photostimulation protocols at each frequency band (*Figure 7E and F*). A test day includes a sequence of a target session with no photostimulation (no stim), and target sessions with 4 Hz, 10 Hz, and 40 Hz photostimulation. The order of sessions was similar in all mice. In all the sessions, photostimulation was applied for 2 s every 30 s and overall changes in the animal's social interactions throughout the session were examined. In the mice expressing ChR2-YFP, 40 Hz photostimulation significantly increased occupancy time in the IZ, compared with no photostimulation (*Figure 7F*; n=7 mice; $t_6$ = 3.46, p=0.040, paired *t*-test followed by Bonferroni correction), whereas no significant changes were observed by 4 Hz and 10 Hz photostimulation (4 Hz: $t_6$=0.56, p>0.99; 10 Hz: $t_6$=0.25, p>0.99, paired *t*-test followed by Bonferroni correction). There were no significant changes in total travel distance by these photostimulation conditions (*Figure 7F*; n=7 mice; p>0.05 versus no stim, paired *t*-test followed by Bonferroni correction). On the other hand, in the mice expressing YFP alone, no significant effects were observed in any photostimulation conditions (*Figure 7G*; n=5 mice; 4 Hz: $t_4$=2.05, p=0.87; 10 Hz: $t_4$=2.16, p=0.83; 40 Hz: $t_4$=2.51, p=0.72, paired *t*-test followed by Bonferroni correction). These results from control experiments exclude the possibility that the increase in IZ occupancy by the 40 Hz photostimulation is simply due to an effect of elapsed time in the test condition. Overall, these results demonstrate that optogenetic inductions of a decrease in dmPFC-BLA 4–7 Hz power and an increase in 30–60 Hz power are sufficient to trigger social interaction behavior.

## Discussion

In this study, we compared LFP signals from the dmPFC and BLA during an SI test among wild-type mice, Shank3 KO mice as a model of ASD, and socially defeated mice as a model of depression. Power spectrum analyses revealed that all mouse types tested exhibited prominent increases in dmPFC-BLA 4–7 Hz power and decreases in 30–60 Hz power throughout a target session compared with a no target session. The dmPFC 4–7 Hz power increase was prominent when socially deficient mouse models exhibited social avoidance behavior. In contrast, dmPFC 4–7 Hz power was dynamically reduced when wild-type mice exhibited social approach behavior compared with leaving behavior with opposite moving directions. Replicating the social interaction-related LFP changes by oscillation-like optogenetic stimulation of PV interneurons was sufficient to increase social interaction in socially defeated mice.

Our results showed that dmPFC 4–7 Hz power is higher during a target session and further increases occur during social avoidance in socially deficient mice. These results may be explained by facts that mice are inherently nervous and anxious against social interaction in a novel test environment and these mental states correlate with increased dmPFC 4–7 Hz power. Indeed, previous observations reported theta (4–10 Hz) power increases in the mPFC-BLA-ventral hippocampal circuit in anxiogenic environments (*Adhikari et al., 2010*; *Likhtik et al., 2014*; *Padilla-Coreano et al., 2019*). However, our analysis by dividing the test box into several segments could not detect apparent LFP power changes associated with anxiety-related locations.

Our observation of the increased dmPFC 30–60 Hz power in the no target session appears inconsistent with a report by *Liu et al., 2020* showing that mPFC low gamma power is decreased when mice explore an empty cage in a three-chamber test. However, this inconsistency may be reconciled by a difference in conditions of the social interaction tests. In our study, the mice were first subject to a no target session in which no target mice were presented anywhere. On the other hand, the three-chamber test by *Liu et al., 2020* initially contained both a target mouse and an empty cage at the same time in an experimental environment, a condition which is likely to be more similar to the

target session, rather than the no target session, utilized in our study. Therefore, the three-chamber test might already induce overall changes in dmPFC 30–60 Hz power in the test environment, as observed from the target session in our study, which were not detected by *Liu et al., 2020*. In both the three chamber test (*Liu et al., 2020*) and the target session in our study, it is consistent that dmPFC 30–60 Hz (or gamma-range) power increases occur when mice approach a target mouse.

A previous report has demonstrated that PFC 2–7 Hz oscillations entrain coherent activity between the amygdala and ventral tegmental area at the beta band and power changes in these oscillations during stress experiences predicts subsequent depression-like behavior (*Kumar et al., 2014*; *Hultman et al., 2016*). Consistent with their observation that normalization of these interregional oscillations by chemogenetic activation of prefrontal-amygdalar circuit reverses stress-induced behavioral deficits (*Hultman et al., 2016*), our results demonstrated that dynamic changes in dmPFC-BLA oscillations at the similar frequency band during behavior indeed correlate with social behavior and exogenous inductions of such oscillations by optogenetic manipulations temporally induced social interaction behavior. An interesting remaining question is how these dynamically changing oscillations are linked with oscillations and coherent activity found in the other brain regions related to stress vulnerability and susceptibility, termed electome factors (*Hultman et al., 2018*).

Generally, studies utilizing an SI test have quantified the duration during which mice stayed in an IZ (*Golden et al., 2011*; *Venzala et al., 2012*; *Ramaker and Dulawa, 2017*). Our analysis failed to detect pronounced differences in LFP power between periods when mice stayed within and outside the IZ. These results were because animals' behavioral patterns were not homogeneous across time in the IZ, suggesting a need to identify detailed behavioral patterns on a moment-to-moment basis to more accurately evaluate social interaction-related cortical LFP signals. Thus, we specifically extracted social approach behavior that more precisely represents animals' psychological states with increased motivation and/or decreased anxiety than simple time spent in the IZ. Based on this identification of a behavioral pattern, we revealed pronounced reductions in 4–7 Hz power associated with social interactions in the dmPFC-BLA circuit.

Our spike analysis confirmed that dmPFC interneurons exhibited spike patterns phase-locked to 4–7 Hz oscillations. Many previous studies have established that PV interneurons are a crucial cell type for generating cortical low gamma-range (20–60 Hz) oscillations (*Whittington et al., 1995*; *Bartos et al., 2007*; *Cardin et al., 2009*; *Sohal et al., 2009*; *Buzsáki and Wang, 2012*; *Nakamura et al., 2015*; *Cao et al., 2018*; *Liu et al., 2020*). The PV-interneuron-mediated gamma oscillation in the PFC is enhanced during social interaction (*Liu et al., 2020*) or attenuated in autism mouse models with social deficits (*Cao et al., 2018*), highlighting the importance of PFC PV interneurons in the expression of social behavior (*Han et al., 2012*; *Xu et al., 2019*) through the regulation of low gamma oscillations (*Cao et al., 2018*; *Liu et al., 2020*). These studies utilized a protocol with 40 Hz photostimulation of PV interneurons with the aim of increasing PFC LFP power at the corresponding frequency (i.e. low gamma) band (*Cao et al., 2018*; *Liu et al., 2020*). Notably, in our study, we initially sought to determine a protocol that could reduce dmPFC 4–7 Hz power to mimic the LFP power changes observed during approach behavior and eventually found that 40 Hz photostimulation selective to PV interneurons was optimal to meet this technical requirement, resulting in a decrease in dmPFC 4–7 Hz and an increase in 40 Hz power. The mechanisms underlying these reciprocal power changes are possibly mediated by a stimulation-induced interference of the entrainment of PV interneuronal spikes by a 4–7 Hz oscillation. Our study demonstrated that such optogenetic manipulations of PV interneurons extend to the BLA circuit and were sufficient to restore social interaction behavior that was reduced in social defeat stress-induced depression mouse models, similar to autism mouse models reported previously (*Cao et al., 2018*). Here, we note that we did not confirm that the phase-locked dmPFC interneurons observed in this study corresponded with PV-positive interneurons. To address these issues, further confirmation is necessary using techniques to identify cell types of recorded neurons such as optogenetic tagging (*Liu et al., 2020*).

Accumulating evidence suggests that disruptions in E/I balances are crucial factors contributing to social behavior deficits in autism-like mouse models (*Rubenstein and Merzenich, 2003*; *Helmeke et al., 2008*; *Yizhar et al., 2011*; *Selimbeyoglu et al., 2017*). From the aspect of neuronal networks, our results add to a growing body of evidence demonstrating that dynamic changes in LFP oscillatory power regulated by inhibitory neuronal networks at appropriate timings are crucial for social behavior. The neurophysiological signatures found in our study may be helpful for a unified mechanistic

understanding of the cellular-based mechanisms and network-based mechanisms underlying sociality and for identifying ASD-related and stress-induced pathophysiology that may lead to the amelioration of social behavior deficits.

## Materials and methods

**Key resources table**

| Reagent type (species) or resource | Designation | Source or reference | Identifiers | Additional information |
|---|---|---|---|---|
| Strain, strain background (*Mus musculus*, male) | C57BL/6 J | SLC | Jax:000664 | |
| Strain, strain background (*Mus musculus*, male) | ICR | SLC | Jax:009122 | |
| Genetic reagent (*M. musculus*) | PV-Cre | The Jackson laboratory | Jax:008069 | |
| Genetic reagent (*M. musculus*) | Shank3KO | PMID:21423165 | | |
| Chemical compound, drug | Isoflurane | Pfizer Inc. | RRID: AB_2734716 | |
| Chemical compound, drug | Phosphate-buffered saline | FUJIFILM Wako Pure Chemical Corporation | Cat# 166-23,555 | |
| Chemical compound, drug | Paraformaldehyde | Sigma-Aldrich | CAS No. 30525-89-4 | |
| Chemical compound, drug | Cresyl violet | Sigma-Aldrich | CAS No. 10510-54-0 | |
| Software, algorithm | Fiji | Fiji is just ImageJ, NIH (https://imagej.net/Fiji) | Fiji, RRID:SCR_002285 | |
| Software, algorithm | Matlab | Mathworks | RRID:SCR_001622 | Version R2020 |
| Other | AAV5-EF1a-DIO-eYFP | UNC Vector Core | In-Stock AAV Vectors –Dr. Karl Deisseroth, 100 ul Aliquots | 1.0×1,013 vg/ml |
| Other | AAV5-EF1a-DIO-ChR2-eYFP | UNC Vector Core | In-Stock AAV Vectors –Dr. Karl Deisseroth, 100 ul Aliquots | 1.0×1,013 vg/ml |

### Animals

All experiments were performed with the approval of the Experimental Ethics Committee at the University of Tokyo (approval number: P29-7 and P29-14) and according to the NIH guidelines for the care and use of mice.

Male C57BL/6 J wild-type mice (8–10 weeks old) with preoperative weights of 20–30 g were used in this study. All the wild-type mice were purchased from SLC (Shizuoka, Japan). Shank3 KO mice were provided by Shionogi & Co., Ltd. and 7 Shank3 KO mice were used for electrophysiological recordings at 8–10 weeks old. PV-IRES-Cre (PV-Cre) mice were obtained from Jackson Laboratory (Jax: 008069; 129P2-Pvalbtm1(cre)Arbr/J) and a subset of PV-Cre mice were subject to social defeat stress at 8–10 weeks old and then used for electrophysiological recordings. The animals were housed and maintained on a 12 h light/12 h dark schedule with lights off at 7:00 AM.

The Shank3 KO mouse was designed with reference to the report by *Peça et al., 2011* with slight modifications (*Peça et al., 2011*). Briefly, a zinc-finger nuclease (ZFN) mRNA (Merck) was microinjected into the pronucleus of fertilized eggs of C57BL/6JJcl mice. The ZFN targets the following sequence of exon13 in the Shank3 gene; TGCTCCCCGCAGAAACcagagaGGACCGGACGAAGCG. The Shank3 deficient founder mice harboring 10 bases deletion in exon13 were identified by genome sequencing and inbreeded to produce homozygous deficient mice. The datasets obtained from Shank3 KO mice were compared with those obtained from the wild-type mice, not littermates of Shank3 KO mice.

### Social defeat

Mice were exposed to chronic social defeat stress as previously described (*Golden et al., 2011*; *Venzala et al., 2012*; *Ramaker and Dulawa, 2017*; *Abe et al., 2019*). At least 1 week before beginning the social defeat experiment, all resident CD-1 mice (SLC, Shizuoka, Japan) more than 13 weeks of age were singly housed on one side of a home cage (termed the 'resident area'; 42.5 cm × 26.6 cm×15.5 cm). The cage was divided into two identical halves by a transparent plexiglas partition (0.5 cm × 41.8 cm×16.5 cm) with perforated holes, each with a diameter of 10 mm. The bedding in

the resident area was left unchanged during the preoperative period. First, resident CD-1 mice were screened for social defeat experiments by introducing an intruder C57/BL6J mouse that was specifically used for screening into the home cage during three 3 min sessions on 3 subsequent days. Each session included a different intruder mouse. CD-1 mice were selected as aggressors in subsequent experiments based on three criteria: during the three 3 min sessions, (1) the mouse attacked in at least two consecutive sessions, (2) the latency to initial aggression was less than 60 s, and (3) the above two criteria were met for at least 2 consecutive days out of 3 test days. After screening, an experimental intruder mouse was exposed to social defeat stress by introducing it into the resident area for a 7–10 min interaction. The interaction period was immediately terminated if the intruder mouse had a wound and bleeding due to the attack, resulting in interaction periods of 7–10 min. After the physical contact, the intruder mouse was transferred across the partition and placed in the opposite compartment of the second resident home cage for the following 24 h; this allowed the intruder mouse to have sensory contact with the resident mouse without physical contact (*Golden et al., 2011*). Over the following 10 days period, the intruder mouse was exposed to a new resident mouse so that the animals did not habituate the same residents.

Defeated control mice were pair housed in the same cage with one mouse per side of the same transparent partition with perforated holes, but they did not experience physical contact with each other (*Golden et al., 2011*).

## Surgical procedures

A single surgery was performed in each mouse for either (i) implantation of a tetrode assembly or (ii) implantation of a tetrode assembly and optic fibers followed by injection of a virus vector. During all surgeries, the animals were anesthetized with isoflurane gas (1–3%), and circular craniotomies were made using a high-speed drill at the indicated coordinates. (i) For LFP recordings without spikes, an electrode assembly that consisted of 3 and 4 immobile tetrodes was stereotaxically implanted above the dmPFC (2.00 mm anterior and 0.50 mm lateral to bregma) at a depth of 1.40 mm and the BLA (0.80 mm posterior and 3.00 mm lateral to bregma) at a depth of 4.40 mm, respectively. For eight wild-type mice and six defeated mice, electrodes were targeted for the dmPFC only. For the other six wild-type mice, five defeated mice, and seven Shank3 KO mice, electrodes were targeted for both the dmPFC and BLA. The tetrodes were constructed from 17-μm-wide polyimide-coated platinum-iridium (90/10%) wires (California Fine Wire), and the electrode tips were plated with platinum to lower the electrode impedances to 200–250 kΩ. Stainless steel screws were implanted on the skull and attached to the cerebellar surface to serve as ground/reference electrodes. For spike recordings, an electrode assembly that consisted of 6 independently movable tetrodes was stereotaxically implanted above the mPFC (1.94 mm anterior and 0.83 mm lateral to bregma) (*Okada et al., 2017*; *Aoki et al., 2019*; *Nishimura et al., 2021*). (ii) For optogenetic experiments, 300 nl AAV5-EF1a-DIO-eYFP or AAV5-EF1a-DIO-ChR2-eYFP (UNC Vector Core, 1.0×10¹³ vg/ml) was injected into the dmPFC (2.00 mm anterior and 0.50 mm lateral to bregma at a depth of 1.40 mm) over 3 min in a PV-Cre mouse, and an optical fiber (core diameter = 200 μm) was then implanted into the same region. In addition, an electrode assembly for LFP recordings was implanted as described above. Finally, all devices were secured to the skull using stainless steel screws and dental cement. After all surgical procedures were completed, anesthesia was discontinued, and the animals were allowed to spontaneously awaken. Following surgery, each animal was housed in a transparent Plexiglas cage with free access to water and food for more than one week.

For spike recordings, the tetrodes were advanced to the targeted brain regions over a period of at least one week following surgery. The depth of the electrodes was adjusted while the mouse rested in a pot placed on a pedestal. The electrode tips were advanced up to 62.5 μm per day over a period of at least 10 days following surgery. The tetrodes were then settled into the targeted area so that stable recordings were obtained.

## Electrophysiological recording

The mouse was connected to the recording equipment via Cereplex M (blackrock), a digitally programmable amplifier, which was placed close to the animal's head. The output of the headstage was conducted to the Cereplex Direct recording system (blackrock), a data acquisition system, via a lightweight multiwire tether and a commutator. For recording electrophysiological signals, the electrical

interface board of the tetrode assembly was connected to a Cereplex M digital headstage (blackrock microsystems), and the digitized signals were transferred to a Cereplex Direct data acquisition system (blackrock microsystems). Electrical signals were sampled at 2 kHz and low-pass filtered at 500 Hz. The unit activity was amplified and bandpass filtered at 750 Hz to 6 kHz. Spike waveforms above a trigger threshold (50 μV) were time-stamped and recorded at 30 kHz in a time window of 1.6ms. The animal's moment-to-moment position was tracked at 15 Hz using a video camera attached to the ceiling. The frame rate of the movie was downsampled to 3 Hz, and the instantaneous speed of each frame was calculated based on the distance traveled within a frame (~333ms). In the following analyses, video frames with massive optical noise or periods that were not precisely recorded due to temporal breaks of image data processing were excluded. All recordings from a behavioral task were performed once so that all the tasks were novel for the mice and no duplications of samples were thus included.

### Social interaction (SI) test

Social interaction tests were performed inside a dark room with a light intensity of 10 lux in a square-shaped box (39.3 cm × 39.3 cm) enclosed by walls 27 cm in height. A wire-mesh cage (6.5 cm × 10 cm×24 cm) was centered against one wall of the arena during both no target and target sessions. Each social interaction test included two 150 s sessions (separated by an intersession interval of 30 s) without and with the target CD-1 mouse present in the mesh cage, termed no target and target sessions, respectively. In the no target session, a test C57BL/6 J mouse was placed in the box and allowed to freely explore the environment. The C57BL/6 J mouse was then removed from the box. In the 30 s break between sessions, the target CD-1 mouse was introduced into the mesh cage. The design of the cage allowed the animal to fit its snout and paws through the mesh cage but not to escape from the cage. In the target session, the same C57BL/6 J mouse was placed beside the wall opposite to the mesh cage. For optogenetic experiments, the duration of a target session was 10 min. In each session, the time spent in the interaction zone (IZ), a 14.5 cm × 24 cm rectangular area extending 8 cm around the mesh cage. The social interaction (SI) ratio was computed as the ratio of time spent in the interaction zone in the target session to the time spent there in the no target session. Social avoidance zones are defined as 9.0 cm × 9.0 cm square areas projecting from both corner joints opposing the cage.

As control experiments, a C57BL/6 J mouse with a weight of 22 g was used as a target mouse or a plastic toy mouse with a similar size to recorded C57BL/6 J mice was placed in the cage instead of a target mouse.

### Optogenetics

The mice underwent one of the following photostimulation protocols during a target session for 10 min: (1) 40 Hz stimulation with 12.5 ms blue light pulses (472 nm,~3 mW output from fiber) at 40 Hz applied with a periodicity of 30 s, (2) 10 Hz stimulation with 50 ms blue light pulses at 10 Hz applied with a periodicity of 30 s, and (3) 4 Hz stimulation with 125 ms blue light pulses at 4 Hz applied with a periodicity of 30 s. In a representative example shown in *Figure 6B* and the analysis in *Figure 6C and D*, each photostimulation was applied for 10 s.

For behavioral experiments, a test day for each mouse includes a sequence of a target session with no photostimulation (no stim), and target sessions with 4 Hz, 10 Hz, and 40 Hz photostimulation. The order of these sessions was similar in all mice. Each session lasted for 10 min, during which photostimulation was applied for 2 s every 30 s, and overall changes in the animal's social interactions throughout the session were examined.

### Histological analysis to confirm electrode placement or cannula placement

The mice were overdosed with isoflurane, perfused intracardially with 4% paraformaldehyde in phosphate-buffered saline (pH 7.4) and decapitated. After dissection, the brains were fixed overnight in 4% PFA and equilibrated with 20 and 30% sucrose in phosphate-buffered saline for an overnight each. Frozen coronal sections (100 μm) were cut using a microtome, and serial sections were mounted and processed for cresyl violet staining. For cresyl violet staining, the slices were rinsed in water, stained with cresyl violet, and coverslipped with Permount. The positions of all electrodes were confirmed by identifying the corresponding electrode tracks in histological tissue. Of the eight

wild-type mice and six defeated mice in which electrodes were targeted for the dmPFC only, eight and three mice had at least one electrode located in the dmPFC, respectively. Of the six wild-type mice, seven Shank3 KO mice, and five defeated mice in which electrodes were targeted for both the dmPFC and BLA, six, seven, and zero mice had at least one electrode located in the dmPFC, and six, seven, and five mice had at least one electrode located in the BLA, respectively.

## LFP analysis

To compute the time-frequency representation of LFP power, LFP signals were convolved using complex Morlet wavelet transformation by the Matlab at frequencies ranging from 1 to 250 Hz. The absolute LFP power spectrum during each 10 ms time window was calculated. In *Figure 2*, z-scores at each frequency band were computed based on the average and SD of LFP power at each frequency band in an entire period including the no target and target sessions. In *Figures 1F, G and 3*, the ratio of absolute power during an entire period of a target session to that during a no target session at a 4–7 Hz or 30–60 Hz band was computed. In *Figures 4, 5*, z-scores at each frequency band were computed based on the average and SD of LFP power at each frequency band in an entire period of the target session. When data were obtained from multiple electrodes in a mouse, they were averaged to single values within each mouse.

Coherence between two electrodes was computed using a Wavelet coherence and cross-spectrum function by the Matlab with a sampling rate of 200 Hz. For quantification, data were obtained and averaged across all dmPFC-BLA electrode pairs.

The Granger causality spectrum between two electrodes was computed using a Matlab function (https://github.com/SacklerCentre/MVGC1; *Barnett, 2021*) with a sampling rate of 200 Hz. For quantification, data were obtained and averaged across all dmPFC-BLA electrode pairs.

## Definition of social approach and leaving behavior

For each location on the animal's trajectories (bin = 1 s), a cage-oriented moving direction θ was computed as an angle between the animal's moving direction at the location and a straight line connecting the location of the animal and the center of the cage. Angles $\theta$=0° and 180° indicate moving directly toward and away from the center of the cage, respectively. Candidate periods of social approach and leaving behavior were defined when the animals stayed in the half of the box containing the IZ and the cage-oriented moving directions were less than and more than 90°, respectively. If a candidate period lasted for more than 5 s, the first 5 s period was regarded as a social approach and leaving behavior. The periods that were not classified as social approach and leaving behavior were termed the other periods.

## Spike unit analysis

Neurons recorded from all electrodes targeting the dmPFC were included in this analysis. Spike sorting was performed offline using the graphical cluster-cutting software Mclust. Sleep recordings obtained before and after the behavioral paradigms were executed were included in the analysis to assure recording stability throughout the experiment and to identify cells that were silent during behavior. Clustering was performed manually in 2D projections of the multidimensional parameter space (i.e. comparisons between the waveform amplitudes, the peak-to-trough amplitude differences, the waveform energies, and the first principal component coefficient [PC1] of the energy-normalized waveform, each measured on the four channels of each tetrode). Only clusters that could be stably tracked across all behavioral sessions were considered to be the same cells and were included in our analysis. Similar to classification criteria in the PFC reported in previous studies (*Wilson et al., 1994*; *Tierney et al., 2004*; *Homayoun and Moghaddam, 2007*), neurons with baseline spike rates of >10 Hz were classified as putative fast-spiking (FS) interneurons, whereas neurons with baseline spike rates of <10 Hz and spike width of >0.6ms were classified as putative regular-spiking (RS) pyramidal neurons. No priori power analyses were performed to determine sample sizes. Experiments were instead designed to encompass a comparable number of cells as several previous studies of spike-phase computation among prefrontal principal cells (e.g. *Karalis et al., 2016*; *Abe et al., 2019*; *Okonogi and Sasaki, 2021*).

For each cell, the degree of phase locking during a target session was analyzed. For approach or leaving behavior, the phase-spike rate distribution was computed by plotting the firing rate as a

function of the phase of 1–4 Hz, 4–7 Hz, 7–15 Hz, 15–30 Hz, and 30–60 Hz LFP traces, divided into bins of 30° and smoothed with a Gaussian kernel filer with standard deviation of one bin (30°), and a Rayleigh r-value was calculated as mean vector length (MVL).

## Statistical analysis

All data are presented as the mean ± SEM, unless otherwise specified, and were analyzed using Python and MATLAB. For normally-distributed data, individual data points are displayed in addition to sample mean and SEM or presented in the sourcedata.xlsx. For non-normally-distributed data, data are displayed as distributions, with data points presented in the sourcedata.xlsx. For each statistical test, data normality was first determined by the F test, and non-parametric tests applied where appropriate. Comparisons of two-sample data were analyzed by paired t-test and Mann-Whitney U test. Multiple group comparisons were performed by post hoc Bonferroni corrections. The null hypothesis was rejected at the p<0.05 level.

## Acknowledgements

This work was supported by KAKENHI (20H03545; 21H05243; 21K19349; 21H05036) from the Japan Society for the Promotion of Science (JSPS), grants (1041630; JP21zf0127004) from the Japan Agency for Medical Research and Development (AMED), grants from the Japan Science and Technology Agency (JST) (JPMJCR21P1), Lotte Research Promotion Grant, the Uehara Memorial Foundation, and Research Foundation for Opto-Science and Technology to T Sasaki; grants from the JST Exploratory Research for Advanced Technology (JPMJER1801), and Institute for AI and Beyond of the University of Tokyo to Y Ikegaya; and a JSPS Research Fellowship for Young Scientists to N Kuga.

## Additional information

### Funding

| Funder | Grant reference number | Author |
|---|---|---|
| Japan Society for the Promotion of Science | 20H03545 | Takuya Sasaki |
| Japan Agency for Medical Research and Development | 1041630 | Takuya Sasaki |
| Japan Science and Technology Agency | JPMJCR21P1 | Takuya Sasaki |
| Japan Science and Technology Agency | JPMJER1801 | Yuji Ikegaya |
| Japan Agency for Medical Research and Development | JP21zf0127004 | Takuya Sasaki |
| Japan Society for the Promotion of Science | 21K19349 | Takuya Sasaki |
| Japan Society for the Promotion of Science | 21H05243 | Takuya Sasaki |
| Japan Society for the Promotion of Science | 21H05036 | Takuya Sasaki |
| Institute for AI and Beyond of the University of Tokyo | | Yuji Ikegaya |
| JSPS Research Fellowship for Young Scientists | | Nahoko Kuga |

The funders had no role in study design, data collection and interpretation, or the decision to submit the work for publication.

## Author contributions
Nahoko Kuga, Conceptualization, Data curation, Formal analysis, Investigation; Reimi Abe, Kotomi Takano, Investigation; Yuji Ikegaya, Supervision, Writing - review and editing; Takuya Sasaki, Conceptualization, Data curation, Formal analysis, Funding acquisition, Investigation, Supervision, Writing - original draft, Writing - review and editing

## Author ORCIDs
Nahoko Kuga (ID) http://orcid.org/0000-0003-0898-1985
Takuya Sasaki (ID) http://orcid.org/0000-0002-0576-7142

## Ethics
All experiments were performed with the approval of the Experimental Ethics Committee at the University of Tokyo (approval number: P29-7 and P29-14) and according to the NIH guidelines for the care and use of mice.

## Decision letter and Author response
Decision letter https://doi.org/10.7554/eLife.78428.sa1
Author response https://doi.org/10.7554/eLife.78428.sa2

## Additional files

### Supplementary files
• Transparent reporting form

### Data availability
All data generated or analyzed during this study are included in the manuscript and supporting file; Source Data files "Kuga et al 2022 sourcedata.xlsx" have been provided for main and supplementary Figures. Original datasets are provided on Mendeley Data (http://doi.org/10.17632/8yv4k58xhj.1).

The following dataset was generated:

| Author(s) | Year | Dataset title | Dataset URL | Database and Identifier |
|---|---|---|---|---|
| Sasaki T | 2022 | Prefrontal-amygdalar oscillations related to social behavior in mice | https://doi.org/10.17632/8yv4k58xhj.1 | Mendeley Data, 10.17632/8yv4k58xhj.1 |

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
