## [Editor Report]

This manuscript is of broad interest to readers studying the neural basis of sociability, social anxiety, and anxiety-like behaviors. The authors use mouse models to understand how electrical oscillations in two key parts of the brain (prefrontal cortex and amygdala) relate to social behavior.

---

## [Decision Letter]

**Decision letter after peer review:**

[Editors’ note: the authors submitted for reconsideration following the decision after peer review. What follows is the decision letter after the first round of review.]

Thank you for submitting the paper "Prefrontal-amygdalar oscillations related to social interaction behavior in mice" for consideration by *eLife*. Your article has been reviewed by 3 peer reviewers, and the evaluation has been overseen by a Reviewing Editor and a Senior Editor. The following individuals involved in review of your submission have agreed to reveal their identity: Nancy Padilla (Reviewer #1); Audrey Brumback (Reviewer #3).

Comments to the Authors:

We are sorry to say that, after consultation with the reviewers, we have decided that this work will not be considered further for publication by *eLife*.

All reviewers agreed that the topic and results are of interest and have potential impact for the field of social cognition. However, all 3 reviewers pointed out the need for additional experiments with proper control groups would be necessary to appropriately interpret the data. Reviewers also felt there was a need for additional detailed analyses, further clarification/explanation of the methods and more discussion on implications/caveats of the findings.

Given the enthusiasm for this report, if the authors feel they can address each of the issues raised by the reviewers with additional data, analyses, and revised text, we would be willing to consider a revised version of the manuscript. The editors and reviewers all agreed that addressing each of the additional control experiments and analyses suggested by the three reviewers will be critical to a successful revision. If you choose to go this route, we will evaluate your revised submission as a new manuscript editorially before sending to review. We would endeavor, but cannot guarantee that we will be able to secure the same reviewers.

*Reviewer #1:*

Kuga and colleagues utilized two mouse models (Shank3 mutants and socially defeated mice) that show sociability deficits to study associations between 4-7 Hz (lower range of theta band) and 30-60 Hz (low γ) oscillations in dmPFC and BLA and social deficits. They record from dmPFC and BLA in two arenas: one with a novel mouse and an empty arena (open field). They show that on average with the novel mouse there is more 4-7 Hz power in dmPFC and BLA, suggesting an association of 4-7 Hz with the social context. Several metrics suggest that the increase in 4-7 Hz power is associated with social avoidance. Finally, they show that stimulating optogenetically mPFC parvalbumin interneurons at 40 Hz (but not lower frequencies) resulted in lower 4-7 Hz power in dmPFC and BLA and more social exploration. To my knowledge this is the first study linking lower range theta in dmPFC and BLA to social avoidance, which is a strength of the study. Furthermore, the use of multiple models with social deficits strengthens that connection to social avoidance. However, the authors calculate metrics differently in the control group and mouse models making it very difficult to compare controls vs mouse models. In addition, the claim of the authors that dmPFC and BLA 4-7 Hz is linked to social interactions (stated in title and summary) is questionable since there are not enough controls to rule out that anxiety-like behavior in the arena is what is driving the changes.

Clarifications on the following are necessary to support the claims:

1. Ample literature links theta oscillations (including the 4-7 Hz range) to anxiety and fear related behaviors in the dmPFC and BLA, as noted by the authors in the introduction. The authors do the social interaction test in a large arena (same size as an open field) therefore the mice will very likely exhibit anxiety-like behavior (thigmotaxis) which will modulate 4-7 Hz in the dmPFC and BLA. Furthermore, the authors used a larger unfamiliar mouse of a different strain, potentially inducing social anxiety. This all brings up the possibility that the changes reported in this manuscript are driven by anxiety-like behavior rather than social interactions (as its claimed in this manuscript). To clarify this the authors should do all the calculations in the empty arena (no target session) to ensure that none of their effects are driven going towards the periphery or by staying in a corner of the field (what they quantify as social avoidance when target is present). For example, what is the difference between 4-7 Hz power during social avoidance vs staying in the same periphery corner in the no target session. Based on the literature these behaviors on the empty arena will likely lead to modulations in theta frequencies, but possibly the modulations are larger when there is a novel mouse in the arena. However, as presented it's not possible to dissociate the findings with potential changes due to anxiety-like behavior.

2. The authors calculate 4-7 Hz power during social approach vs leaving in the WT mice but in the other two groups they calculate it during time spent in the corner (social avoidance). A central claim of this manuscript (expressed in summary and in lines 216-218) is that the 4-7 Hz patterns are opposite in the social deficit models vs the control. However, the way 4-7 Hz was calculated across groups are not the same, so that claim currently has an important caveat. First, uniformity of calculating metrics will improve the rigor and clarity of the paper. Second, power during both behaviors (social avoidance and social leaving) suggest that 4-7 Hz is associated with not interacting, rather than with social interaction. Assuming that the dissociation from anxiety can be done (point 1), the data support that 4-7 Hz dmPFC and BLA power is associated with social avoidance, not social interactions.

3. This study normalizes LFP signals within animals making it difficult to understand if the patterns reported only hold with relative measurements only or if 4-7 Hz absolute power across groups differs in the distinct conditions. In addition, only two example power spectra, one with a log scale that makes it hard to see 4-7 Hz, are shown in the paper making it difficult to assess the 4-7 Hz peaks across groups and conditions and to understand why 4-7 Hz was selected as a frequency band. This study will highly benefit from including average power spectra for all conditions and groups.

4. Given the optoelectric effect that light can have in electrodes which impacts LFPs (see Cardin et al., 2012), it is important to rule out that the power differences reported in mPFC in Figure 6 are not driven by light presentation. The authors only show data from an opsin group, so there is no control group to address this concern.

5. Given that locomotion can change 4-7 Hz power in the mPFC (see Adhikari et al., 2010) it is important to rule out that the increased in 4-7 Hz power seen in the target session relative to the no target session is not driven by a difference in locomotion.

6. The manuscript should cite and discuss how the current data can be consolidated with this study by Liu et al., 2020 that stimulated mPFC PV interneurons at γ frequencies and sees changes in sociability. However, they report that low γ in the mPFC increases during a social session compared to empty arena, which is opposite from what the authors currently report. https://pubmed.ncbi.nlm.nih.gov/32832654/

Recommendation for the authors:

1. To rule out that the increased in 4-7 Hz power seen in the target session relative to the no target session is not driven by a difference in locomotion, one approach is to see if there is a statistical difference in speed in one session vs the other. Alternatively, if there is a difference in speed, the other approach is to parse the data by speed and only use LFP signals from times in which the animals are moving at the same speed.

2. Currently, it is not described how the z-scoring of the LFP signal is done. This is very important information that should be clearly explained in the methods for rigor and reproducibility.

3. Given that the 2 mouse models are using the same control group data (Figures 2-3), the appropriate approach is to plot the 3 groups together, do an ANOVA then post-hoc comparisons that control for multiple comparisons. Currently, the authors are treating the control groups for the two social deficit models as independent, but they are the same data points. I do not think that grouping them adequately would change the conclusions reported but given the multiple comparisons using the same control group, this will be the most transparent and clear way to report the data.

4. In addition, it should be made clear in the methods the that the same WT group was used as control for both social deficit models, including the rationale. Normally, in social defeat studies control mice experience daily cage changing and handling to control for the daily experience of the defeated mice (see Golden et al., 2011, Krishnan and Nestler 2008 for control details). On the other hand, when using mutant mice, the best controls are wild type littermates.

5. Given that 40 Hz stimulation altered 4-7 Hz power this suggests that theta-γ coupling is occurring, which is a phenomenon that occurs in both mPFC and BLA (see Stujenske et al., 2014). The paper would be improved by adding discussion of this phenomena and quantifying if coupling exists between the γ band and the 4-7 Hz during the social session and if it’s modulated by social avoidance.

6. Finally, given that dmPFC and BLA recordings were simultaneous reporting the coherence and directionality of signaling between dmPFC and BLA and how they are modulated between social vs no social conditions would strengthen the claim that dmPFC and BLA circuit is working together.

7. The methods describe an open field which I assume was used as the no target session, but this is not explicitly stated. If the open field was the no target session, this needs to be clearly stated in the methods.

8. The manuscript should be revised for typos. FS was labeled as PS in line 258

*Reviewer #2:*

The dmPFC and BLA play important roles in regulating social behavior. In this manuscript, Nahoko Kuga et al. investigate how neural activity in the dmPFC and BLA is modulated during social interaction. By recording local field potentials from dmPFC and BLA during social interaction, the authors find that the dmPFC and BLA show modulations of 4-7 Hz and 30-60 Hz oscillations during social interaction. In wild type animals, 4-7 Hz oscillation power increases and 30-60 Hz power decreases during social interaction, and this bidirectional modulation is associated with social avoidance behavior. During social approach, however, 4-7 Hz oscillation power decreases. Interestingly, mouse models with reduced social interactions (Shank3 knockout and socially defeated animals) display a further increase in 4-7 Hz oscillation power during social avoidance behavior compared to wild type animals. These new results are interesting, as they confirm the findings from previous studies that theta and low γ bands are modulated during social interaction and social anxiety and further show that the BLA and dmPFC are modulated in a similar manner.

Several aspects of the manuscript could benefit from additional experiments and controls and further data analysis. While a majority of animals show a positive preference for social targets (Figure 1B), the use of CD1 animals as target animals could lead to more social avoidance, as CD1 animals are substantially larger and may increase more social avoidance behavior than regular C57BL/6J animals. The authors show that 4-7 Hz and 30-60 Hz power is heavily modulated during specific types of social behavior, such as approaching vs. leaving the interaction zone. This suggests that the neural activity changes in Shank3 or socially defeated animals reported in Figures 2 and 3 could simply reflect changes in behavior compositions, as opposed to changes in the underlying neural encoding of these behaviors.

Specific comments:

1. Previous studies have already reported modulation of theta and low γ bands during social interaction (Liu et al. 2020 Science Advances), and that 40 Hz modulation of PV neurons can restore social behavior deficits in mouse models with reduced social interaction (Cao et al. 2018 Neuron). An increase in activity of wide-spiking cells and a decrease in activity of fast-spiking cells have also been reported in social anxiety (Xu et al. 2019 Neuron). Some of these studies also used more sophisticated approaches, such as combining in vivo electrophysiology with opto-tagging to identify specific interneuron cell types. The authors should discuss their new findings in relation to these previous studies.

2. The authors show an increase in 4-7 Hz power and decrease in 30-60 Hz power during social interaction, and that these changes are associated with social avoidance behavior. While a majority of animals show a positive preference for social targets (Figure 1B), the use of CD1 animals as target animals could lead to more social avoidance, as CD1 animals are substantially larger and may increase more social avoidance behavior than regular C57BL/6J animals. The authors could use unfamiliar C57BL/6J mice as target animals. Also, the authors could incorporate an additional control with a toy mouse instead of an empty cage in the experimental setup.

3. The authors show that 4-7 Hz and 30-60 Hz power is heavily modulated during specific types of social behavior, such as approaching vs. leaving the interaction zone. This suggests that the neural activity changes in Shank3 or socially defeated animals reported in Figures 2 and 3 could simply reflect changes in behavior compositions, as opposed to changes in the underlying neural encoding of these behaviors. The authors could perform more in-depth analyses of the behaviors of Shank3 or socially defeated animals using a similar approach to determine whether and how Shank3 or socially defeated animals display different compositions of behavior in their assays. The authors could also investigate the modulations of 4-7 Hz and 30-60 Hz power during these specific behaviors in Shank3 or socially defeated animals.

4. Given that the behavior repertoire is diverse across wild-type mice (Figure 1B), the current work would benefit from examining whether the level of social motivation is related to the magnitude of change of dmPFC 4-7 Hz signal in wild type mice. More specifically, this can be addressed by splitting wild-type mice into subgroups with high and low social motivation, followed by examining if the change of dmPFC 4-7 Hz band in the low social motivation subgroup is higher than that of the high social motivation subgroup, and whether it is comparable to Shank3 and socially defeated groups. Alternatively, correlating percent time of social interaction to the change magnitude of dmPFC 4-7Hz through a simple linear regression model would also help enhance the generalizability of the claim overall.

5. While the authors simultaneously recorded LFP signals from both dmPFC and BLA, the temporal relationship between dmPFC and BLA oscillatory signals was not explored. The impact of the current work can be bolstered through a deeper characterization of their temporal relationship. The simultaneous LFP recording in both regions provide an opportunity to examine the phase relationship of the 4-7 Hz band (and others) during different social epochs (e.g. approaching/leaving the interaction zone) which may shed light on whether PFC-amygdalar communication is involved in social anxiety-related behaviors.

*Reviewer #3:*

Here the authors are using optogenetics combined with in vivo recordings of mouse prefrontal cortex and amygdala during social behavior to test the sufficiency of different frequency bands for those social behaviors. Once establishing how the oscillations change during social behavior in wildtype mice, they identify how these oscillations are similar and different in two mouse models of neuropsychiatric disorders with decreased social interactions. In one of the disease models, they use optogenetics to enhance the power of the oscillations and find that it increases social behavior in that model.

The authors examine modulation of theta and γ power in dorsal medial prefrontal cortex (dmPFC) and basolateral amygdala (BLA) during social exploration behavior. The authors performed simultaneous recordings from the dmPFC and BLA during a social interaction test and found increases in theta power and decreases in γ power compared to the non-social interaction trial. Social defeat animals and Shank3 KO mice had similar alterations in their theta frequency bands compared to control mice. Optogenetic manipulation of γ frequency bands increased social exploration behavior in social defeat mice.

The strengths of the paper include the importance of the question being asked, including the brain regions studied and in studying the potential for convergent biological changes in abnormal behavior arising from different etiologies. Experimentally, a strength is the combination of simultaneous in vivo physiology of two brain regions with optogenetics in awake, behaving mice. These are difficult experiments. A further strength is the nuanced analysis of animal behavior and the measurement of brain oscillations during those phases (i.e. approach vs. leaving).

The wildtype data are intriguing and interesting, as they help fill in the gaps in our understanding of how and when the prefrontal cortex and amygdala activate during social behavior, and how this is similar and different in disease models that display decreased social interactions.

There is an over-reliance on multiple t-tests instead of using more appropriate statistical analyses such as ANOVA.

A barrier to interpreting the results of the disease model work is the lack of appropriate controls. The authors do not use littermate controls for the Shank3 KO vs. wildtype comparisons. Instead, they compare Shank3 KO mice from one line of mice to wildtype mice from another line. It is well-known that different lines of mice can have drastically different behavior and physiology. Second, there are no untreated control animals for any of the social defeat optogenetics experiments, so it is unknown if the effects observed are specific to the disease model or nonspecific effects. There are appropriate eYFP-only controls and no-light controls.

There are multiple recommendations for improvement:

For all figures, I recommend showing individual data points for each animal in addition to the mean +/- SEM. There is no clear reason this is done in some graphs but not others.

Figure S2 – I wonder if performing multiple paired t-test is the correct statistical treatment for the comparisons, since you're dividing the data into multiple different frequency bands in the same animal.

For the data presented in Figure 1 and S2, it is unclear if the powers measured are for the whole trial or only when the animal is in the "interaction zone".

In figure 1E, it is unclear what is being averaged if this is a single mouse. Data are presented as the mean +/- SEM, but it’s unclear what samples are used to calculate this.

Consider adding a horizontal line to Figures 1 F and G showing where Z = 0. It is unclear what is being used as the baseline for the Z score.

N=14 mice for the dmPFC recordings and N=6 for the BLA recordings. If all recordings were done with both locations targeted but fewer BLA animals are presented, please indicate why (presumably because many of the BLA recordings were not targeted correctly).

The order of data presentation in Figure 2 is confusing because it leads the reader to think that the data presented in 2C and D have something to do with the drawing presented in 2B. Based on the text, the drawing in 2B only pertains apparently to the data presented in 2E and F.

P7, L134-136 – it is unclear if this interpretation is based on the KO mice having lower social behavior than WT mice as a group, or on a per-animal basis.

The Shank3 KO mice appear to be tested without a WT littermate control group. The WT mice used in the initial part of the study are a separate cohort of mice from a different lineage and are then used for comparison with the Shank3 KO mice.

P7, L137-138 – it is unclear whether the Shank3 mice spending "substantial" time in the avoidance zone is different from WT mice. For the associated Figure 2EandF, there is no point of comparison with wildtype mice to know if control mice also display similar changes in theta power when in the avoidance zone.

The authors find increases in theta power during social trials and decreases in theta power during social avoidance in Shank3 KO and social defeat animals.

The analysis of the spiking data is confusing. For example, p12, L248 says neurons were excluded from further analysis if they didn't show a preference in phase locking for approaching vs. leaving. Then the authors discuss how the remaining FS neurons showed higher theta phase locking during leaving. The confusing part is that then they discuss how RS neurons "did not show such significance". By definition, the neurons should have been excluded from this very analysis if they didn't show a significant difference between leaving and approaching.

For Figure S4, I am concerned that doing multiple paired t-tests is not the appropriate statistics for these multiple frequency bands. I would recommend at least an ANOVA.

The figures are aesthetically pleasing.

---

## [Author Response]

[Editors’ note: the authors resubmitted a revised version of the paper for consideration. What follows is the authors’ response to the first round of review.]

Reviewer #1:Kuga and colleagues utilized two mouse models (Shank3 mutants and socially defeated mice) that show sociability deficits to study associations between 4-7 Hz (lower range of theta band) and 30-60 Hz (low γ) oscillations in dmPFC and BLA and social deficits. They record from dmPFC and BLA in two arenas: one with a novel mouse and an empty arena (open field). They show that on average with the novel mouse there is more 4-7 Hz power in dmPFC and BLA, suggesting an association of 4-7 Hz with the social context. Several metrics suggest that the increase in 4-7 Hz power is associated with social avoidance. Finally, they show that stimulating optogenetically mPFC parvalbumin interneurons at 40 Hz (but not lower frequencies) resulted in lower 4-7 Hz power in dmPFC and BLA and more social exploration. To my knowledge this is the first study linking lower range theta in dmPFC and BLA to social avoidance, which is a strength of the study. Furthermore, the use of multiple models with social deficits strengthens that connection to social avoidance. However, the authors calculate metrics differently in the control group and mouse models making it very difficult to compare controls vs mouse models. In addition, the claim of the authors that dmPFC and BLA 4-7 Hz is linked to social interactions (stated in title and summary) is questionable since there are not enough controls to rule out that anxiety-like behavior in the arena is what is driving the changes.Clarifications on the following are necessary to support the claims:1.1. Ample literature links theta oscillations (including the 4-7 Hz range) to anxiety and fear related behaviors in the dmPFC and BLA, as noted by the authors in the introduction. The authors do the social interaction test in a large arena (same size as an open field) therefore the mice will very likely exhibit anxiety-like behavior (thigmotaxis) which will modulate 4-7 Hz in the dmPFC and BLA.

We appreciate this crucial comment. To address these concerns, we performed additional experiments and analyses as below.

1.2. Furthermore, the authors used a larger unfamiliar mouse of a different strain, potentially inducing social anxiety. This all brings up the possibility that the changes reported in this manuscript are driven by anxiety-like behavior rather than social interactions (as its claimed in this manuscript).

As the reviewer pointed out, we utilized a larger CD-1 mouse as a target mouse in the test, which may induce more anxiety-related or fearful behavior. To reduce the effects of these emotional factors, we performed additional tests using an unfamiliar C57BL/6J mouse with a similar body size as a target mouse (new Figure 1J). In this condition, we found significant changes in dmPFC 4–7 Hz and 30–60 Hz power in the target session, similar to the results from the larger CD-1 mouse as shown in Figure 1F. This result confirms that the dmPFC power changes found in this study are induced by social behavior and are less associated with emotional components such as anxiety.

In addition, as a control test with no social behavior, we performed additional experiments by placing a plastic toy mouse in the cage as a novel object instead of a real mouse (new Figure 1K). In this condition, we found no significant dmPFC 4–7 Hz and 30–60 Hz power changes between the two sessions. This result confirms that the dmPFC power changes are induced specifically in a condition where mice exhibit social behavior while they are less associated with a novelty.

These results are presented in Figure 1J and 1K and described in the Results part (Line 168-184).

1.3. To clarify this the authors should do all the calculations in the empty arena (no target session) to ensure that none of their effects are driven going towards the periphery or by staying in a corner of the field (what they quantify as social avoidance when target is present). For example, what is the difference between 4-7 Hz power during social avoidance vs staying in the same periphery corner in the no target session. Based on the literature these behaviors on the empty arena will likely lead to modulations in theta frequencies, but possibly the modulations are larger when there is a novel mouse in the arena. However, as presented it's not possible to dissociate the findings with potential changes due to anxiety-like behavior.

We appreciate this great idea. To test whether the observed LFP power changes in the target session may be explained by increased anxiety, we performed additional analyses and created new Figure 2.

Here, the SI test field was divided into social avoidance zones, peripheral areas, and a center area (Figure 2A). In both the no target and target sessions, no significant differences in dmPFC and BLA 4–7 Hz and 30–60 Hz LFP power were observed among these areas (Figure 2B and 2C). These results demonstrate that anxiety-related behavior alone does not induce prominent changes in dmPFC and BLA 4–7 Hz and 30–60 Hz LFP power found in this study. In the Results part, we added a new paragraph to explain these results including statistics and their backgrounds (for more detail, please see Line 185214).

Based on these new findings, we rewrote the Discussion part related to anxiety induced LFP patterns (Line 471-479).

2.1. The authors calculate 4-7 Hz power during social approach vs leaving in the WT mice but in the other two groups they calculate it during time spent in the corner (social avoidance). A central claim of this manuscript (expressed in summary and in lines 216-218) is that the 4-7 Hz patterns are opposite in the social deficit models vs the control. However, the way 4-7 Hz was calculated across groups are not the same, so that claim currently has an important caveat. Firstly, uniformity of calculating metrics will improve the rigor and clarity of the paper.

We appreciate this constructive suggestion.

First, to consistently apply the same analyses across the three mouse types, we changed our analysis (from z-scores to %changes) to compare differences between the no target and target sessions in the wild-type mice in Figure 1F and 1G. These results are now fairly comparable to the Shank3 KO and defeated mice in Figure 3B and 3C.

In addition, similar to socially deficient mice, we newly analyzed how social avoidance behavior affects LFP patterns in the wild-type mice in Figure 4A and found no significant changes in LFP power by social avoidance behavior. This analysis is now comparable to the analyses applied to Shank3 KO and defeated mice. Overall, the new Figure 4 demonstrated clear differences across wild-type mice, Shank3 KO mice, and defeated mice.

In Figure 5 (not changed from the previous manuscript), we analyzed how social approach behavior affects LFP patterns in wild-type mice. While Shank3 KO mice exhibited less social interaction, they still showed substantial approach behavior (Supplementary Figure 4A). We thus applied the same analysis to Shank3 KO mice by extracting approach behavior, similar to the wild-type mice, and found that they did not show significant differences in dmPFC 4–7 Hz and 30–60 Hz power between approach and leaving behavior (Supplementary Figure 4D). This result suggests that social behavior-related dmPFC activity is not properly regulated in socially deficient mice. These results are added in the Results part (Line 335-340).

In defeated mice, this analysis was not applicable because most of them did not show sufficient social approach trials (Figure 3A and Supplementary Figure 4A).

Taken together, we consider that these new sets of additional analyses that are consistently applied across mouse groups and the reorganization of these figures demonstrate comparable differences across the mouse groups.

2.2. Secondly, power during both behaviors (social avoidance and social leaving) suggest that 4-7 Hz is associated with not interacting, rather than with social interaction. Assuming that the dissociation from anxiety can be done (point 1), the data support that 4-7 Hz dmPFC and BLA power is associated with social avoidance, not social interactions.

This is also a crucial point. While our results support that dmPFC 4–7 Hz oscillations dynamically change with social behavior, rather than anxiety as stated above, we could not fully conclude whether they are more strongly related to social interaction or social avoidance. Therefore, throughout the manuscript, we weakened our statement related to “social interaction” and simply described “social behavior”. For example, in the Title and Discussion part, we replaced our statement “social interaction behavior” to “social behavior”.

3. This study normalizes LFP signals within animals making it difficult to understand if the patterns reported only hold with relative measurements only or if 4-7 Hz absolute power across groups differs in the distinct conditions. In addition, only two example power spectra, one with a log scale that makes it hard to see 4-7 Hz, are shown in the paper making it difficult to assess the 4-7 Hz peaks across groups and conditions and to understand why 4-7 Hz was selected as a frequency band. This study will highly benefit from including average power spectra for all conditions and groups.

We appreciate this helpful suggestion. Indeed, the previous figure was hard to understand.

First, we presented new LFP spectrums “without normalization” in Figure 1E top. These spectrums were obtained by averaging over all mice. In addition, we presented all datasets obtained from individual mice in Supplementary Figure 2A. As shown in this figure, absolute LFP power spectrum was variable across individual mice, presumably due to differences in recording conditions (e.g. electrode impedance, recording sites, or individual differences of baseline). In spite of these individual differences, their averages exhibited differences between the two sessions at relatively lower (below 10 Hz) and higher (tens of Hz) frequency bands (Figure 1E, top).

To further emphasize these differences, we newly created a spectrum representing ratios of LFP power in the target session relative to that in the no target session (Figure 1E, bottom). The spectrum more clearly visualized a significant increase and decrease in dmPFC power at a frequency band of 4–7 Hz and 30–60 Hz, respectively, consistent with our statistical results (Figure 1F). Moreover, Supplementary Figure 3A and 3B demonstrate that these LFP power changes were not observed outside the 4–7 Hz and 30–60 Hz bands.

These results are now added in the Results part (Line 102-125).

4. Given the optoelectric effect that light can have in electrodes which impacts LFPs (see Cardin et al., 2012), it is important to rule out that the power differences reported in mPFC in Figure 6 are not driven by light presentation. The authors only show data from an opsin group, so there is no control group to address this concern.

To address this concern, we performed additional experiments by using control PVCre mice injected with AAV5-EF1a-DIO-eYFP (YFP expression only). We confirmed that photostimulation to these control mice did not significantly change dmPFC power. These results are now presented in Figure 7D and described in the Results part (Line 433-436).

5. Given that locomotion can change 4-7 Hz power in the mPFC (see Adhikari et al., 2010) it is important to rule out that the increased in 4-7 Hz power seen in the target session relative to the no target session is not driven by a difference in locomotion.

We appreciate this crucial suggestion. To address this comment, we first compared moving speed between the two sessions and found that overall moving speed was significantly higher in the no target session than the target session. This result is now presented in Supplementary Figure 2D.

To confirm whether locomotion affects LFP power, we next compared LFP power between running periods with a moving speed of more than 5 cm/s and stop periods with a moving speed of less than 1 cm/s, which occupied 20.2% and 40.6%of entire recording periods, respectively (Supplementary Figure 2E and 2F). Both in the dmPFC and BLA, 4–7 Hz power during stop periods was significantly higher than that during running periods, suggesting that locomotion affects these LFP power changes.

Therefore, to address the reviewer’s concern, we performed the same analysis (to Figure 1F) by specifically extracting these behavioral patterns with high and low moving speed in Supplementary Figure 2G and 2H. Interestingly, the significant changes in dmPFC LFP power between the two sessions were specifically observed during stop periods, but not running periods. These results confirm that, while 4–7 Hz and 30–60 Hz power in the dmPFC and BLA is higher and lower, respectively, as moving speed is lower, the LFP power changes found in this study were still prominent in the target session when mice stopped. These results are now described in the Results part (Line 146-167).

6. The manuscript should cite and discuss how the current data can be consolidated with this study by Liu et al., 2020 that stimulated mPFC PV interneurons at γ frequencies and sees changes in sociability. However, they report that low γ in the mPFC increases during a social session compared to empty arena, which is opposite from what the authors currently report. https://pubmed.ncbi.nlm.nih.gov/32832654/

We appreciate this comment. We made an additional paragraph In the Discussion part and described possible explanations as follows:

“Our observation of the increased dmPFC 30–60 Hz power in the no target session appears inconsistent with a report by Liu et al. (2020) showing that mPFC low γ power is decreased when mice explore an empty cage in a three-chamber test. However, this inconsistency may be reconciled by a difference in conditions of the social interaction tests. In our study, the mice were first subject to a no target session in which no target mice were presented anywhere. On the other hand, the three-chamber test by Liu et al. (2020) initially contained both a target mouse and an empty cage at the same time in an experimental environment, a condition which is likely to be more similar to the target session, rather than the no target session, utilized in our study. Therefore, the three-chamber test itself might already induce overall changes in dmPFC 30–60 Hz power in the test environment, as observed from the target session in our study, which were not detected by Liu et al. (2020). In both the three chamber test (Liu et al., 2020) and the target session in our study, it is consistent that dmPFC 30–60 Hz (or γ-range) power increases occur when mice approach a target mouse.” (Line 480-493)

7. To rule out that the increased in 4-7 Hz power seen in the target session relative to the no target session is not driven by a difference in locomotion, one approach is to see if there is a statistical difference in speed in one session vs the other. Alternatively, if there is a difference in speed, the other approach is to parse the data by speed and only use LFP signals from times in which the animals are moving at the same speed.

We appreciate this constructive suggestion. Our answer to this comment is summarized in Comment #5 above.

8. Currently, it is not described how the z-scoring of the LFP signal is done. This is very important information that should be clearly explained in the methods for rigor and reproducibility.

First, in Figure 1F and 1G (one of our main results), we changed our analysis (from z-scores to %changes) to be consistent with Shank3 KO mice and defeated mice as in Figure 3B and 3C.

As for Figure 2B and 2C, we added a following explanation in the Results part (Line 197-200): “To compare relative changes in LFP power across behavior and sessions, LFP power at each frequency band was z-scored based on the average and SD of LFP power at each frequency band in an entire period including the no target and target sessions.” In addition, we added a following explanation in the Methods part (Line 722-724): “In Figure 2, z-scores at each frequency band were computed based on the average and SD of LFP power at each frequency band in an entire period including the no target and target sessions.”

As for Figure 4 and 5, we added a following explanation in the Methods part (Line 726-728): “In Figure 4 and 5, z-scores at each frequency band were computed based on the average and SD of LFP power at each frequency band in an entire period of the target session.”

For LFP power analysis, it is not easy to determine how we take a baseline to normalize datasets as behavioral patterns themselves differ across animals. Here, we tested several periods (i.e. resting periods only, approach periods, and avoidance periods) as baselines and confirmed statistical results became almost similar. After all, for simplicity, we took entire session periods as a baseline.

9. Given that the 2 mouse models are using the same control group data (Figures 2-3), the appropriate approach is to plot the 3 groups together, do an ANOVA then post-hoc comparisons that control for multiple comparisons. Currently, the authors are treating the control groups for the two social deficit models as independent, but they are the same data points. I do not think that grouping them adequately would change the conclusions reported but given the multiple comparisons using the same control group, this will be the most transparent and clear way to report the data.

According to the suggestion, we combined the comparisons across WT mice, Shank3 KO mice (previous Figure 2), defeated mice (previous Figure 3) in one Figure (new Figure 3). In all analyses, we now applied ANOVA and Bonferroni correction for multiple comparisons (all statistical values are described in the manuscript (please see the first paragraph of “Increases in dmPFC 4–7 Hz power during social avoidance in socially deficient mouse models”)). These corrections did not alter our results from the previous manuscript.

10. In addition, it should be made clear in the methods the that the same WT group was used as control for both social deficit models, including the rationale. Normally, in social defeat studies control mice experience daily cage changing and handling to control for the daily experience of the defeated mice (see Golden et al., 2011, Krishnan and Nestler 2008 for control details). On the other hand, when using mutant mice, the best controls are wild type littermates.

As control experiments to compare with socially defeated mice, we performed additional experiments with defeated control mice that were pair housed in the same cage with aggressor mice but not subject to physical contact, as the reviewer suggested. Results from the comparison between these defeated control mice and defeated mice are now presented in Supplementary Figure 4F, confirming that defeated control mice, as well as wild-type mice (Figure 3B), exhibit significantly lower increases in dmPFC 4-7 Hz power, compared with defeated mice. These results are now described in the Results part (Line 271-275).

As the reviewer suggested, utilizing the same WT littermates to Shank3 KO mice is ideal as control mice. Unfortunately, due to housing problems in our vivarium, these mice are no more available.

We added this limitation in the Methods part as follows: “The datasets obtained from Shank3 KO mice were compared with those obtained from the wild-type mice, not littermates of Shank3 KO mice.” (Line 575-577)

11. Given that 40 Hz stimulation altered 4-7 Hz power this suggests that theta-γ coupling is occurring, which is a phenomenon that occurs in both mPFC and BLA (see Stujenske et al., 2014). The paper would be improved by adding discussion of this phenomena and quantifying if coupling exists between the γ band and the 4-7 Hz during the social session and if it’s modulated by social avoidance.

We appreciate this interesting recommendation.

As we did not present this data in the first version of the manuscript, we have indeed tested phase-amplitude coupling analyses for LFP signals during a variety of states (e.g. entire target session, social avoidance, or social interaction). After all, we could not observe pronounced coupling between the 4–7 Hz and 30–60 Hz frequency bands in both the dmPFC and BLA LFP traces in any behavioral conditions. In the revised version, we thus only presented a representative result in Supplementary Figure 2B and just briefly mentioned this result at Line 125-129.

While we initially expected prominent coupling as shown by Stujenske et al. (2014), our result was not significant. This result is likely because the directions of changes in the 4–7 Hz and 30–60 Hz power were opposite, as claimed in this paper.

12. Finally, given that dmPFC and BLA recordings were simultaneous reporting the coherence and directionality of signaling between dmPFC and BLA and how they are modulated between social vs no social conditions would strengthen the claim that dmPFC and BLA circuit is working together.

We appreciate this idea. This computation was possible as we obtained simultaneous recordings from the dmPFC and BLA from six wild-type mice and seven Shank3 KO mice.

First, we computed coherence and Granger causality from all no target and target sessions in Figure 1H and 1I. Similar to the power changes in Figure 1F and 1G, dmPFCBLA coherence at the 4–7 Hz band in the target session was significantly higher than that in the no target session, confirming the coordination of dmPFC-BLA 4–7 Hz. Furthermore, the Granger causality spectrum exhibits a significantly higher Granger causality index at the 4–7 Hz band for the direction from the dmPFC to BLA than that for the direction from the BLA to the dmPFC, possibly representing the preferential projection of the dmPFC to the BLA (Gabbott et al., 2005; Bukalo et al., 2015). These results are added in the Results part (Line 129-138).

Second, we applied the same analysis to social avoidance behavior in Shank3 KO mice. However, dmPFC-BLA coherence and the directionality at the 4–7 Hz band were not prominent during social avoidance behavior. Their exact mechanisms are unknown but we presented the data in Supplementary Figure 4B and 4C and described these results at Line 252-257.

Third, we applied the same analysis to approach and leaving behavior in wild-type mice. Consistent with the LFP power changes (Figure 5F), dmPFC-BLA coherence at the 4– 7 Hz band during leaving behavior was significantly higher than that during approach behavior (Figure 5F, right). Moreover, the Granger causality index at the 4–7 Hz band in the dmPFC-BLA direction was significantly higher than that in the BLA-dmPFC direction during approach leaving behavior, but not during approach behavior (Figure 5G). These results suggest that functional information transfer at the 4–7 Hz band from the dmPFC to the BLA is lowered during social approach behavior, compared with leaving behavior. These results are added in the Results part (Line 326-335).

13. The methods describe an open field which I assume was used as the no target session, but this is not explicitly stated. If the open field was the no target session, this needs to be clearly stated in the methods.

In the no target session in our study, an empty cage was placed in the square test box. To our understanding, as the term “open field” refers a field with completely no objects, our test environment was not perfectly similar to an open field. We described the details of this method in the manuscript as follows:

Methods part (Line 667-668): “A wire-mesh cage (6.5 cm × 10 cm × 24 cm) was centered against one wall of the arena during both no target and target sessions.”

Results part (Line 91-93): “C57BL/6J mice were tested in a conventional SI test in which

they freely interacted with an empty cage and the same cage containing a target CD-1 mouse for 150 s, termed a no target and a target session, respectively.”

In addition, we removed a term “test field” throughout the manuscript.

14. The manuscript should be revised for typos. FS was labeled as PS in line 258

Corrected (Line 385). Thank you for finding this typo.

Reviewer #2:The dmPFC and BLA play important roles in regulating social behavior. In this manuscript, Nahoko Kuga et al. investigate how neural activity in the dmPFC and BLA is modulated during social interaction. By recording local field potentials from dmPFC and BLA during social interaction, the authors find that the dmPFC and BLA show modulations of 4-7 Hz and 30-60 Hz oscillations during social interaction. In wild type animals, 4-7 Hz oscillation power increases and 30-60 Hz power decreases during social interaction, and this bidirectional modulation is associated with social avoidance behavior. During social approach, however, 4-7 Hz oscillation power decreases. Interestingly, mouse models with reduced social interactions (Shank3 knockout and socially defeated animals) display a further increase in 4-7 Hz oscillation power during social avoidance behavior compared to wild type animals. These new results are interesting, as they confirm the findings from previous studies that theta and low γ bands are modulated during social interaction and social anxiety and further show that the BLA and dmPFC are modulated in a similar manner.Several aspects of the manuscript could benefit from additional experiments and controls and further data analysis. While a majority of animals show a positive preference for social targets (Figure 1B), the use of CD1 animals as target animals could lead to more social avoidance, as CD1 animals are substantially larger and may increase more social avoidance behavior than regular C57BL/6J animals. The authors show that 4-7 Hz and 30-60 Hz power is heavily modulated during specific types of social behavior, such as approaching vs. leaving the interaction zone. This suggests that the neural activity changes in Shank3 or socially defeated animals reported in Figures 2 and 3 could simply reflect changes in behavior compositions, as opposed to changes in the underlying neural encoding of these behaviors.Specific comments:1. Previous studies have already reported modulation of theta and low γ bands during social interaction (Liu et al. 2020 Science Advances), and that 40 Hz modulation of PV neurons can restore social behavior deficits in mouse models with reduced social interaction (Cao et al. 2018 Neuron). An increase in activity of wide-spiking cells and a decrease in activity of fast-spiking cells have also been reported in social anxiety (Xu et al. 2019 Neuron). Some of these studies also used more sophisticated approaches, such as combining in vivo electrophysiology with opto-tagging to identify specific interneuron cell types. The authors should discuss their new findings in relation to these previous studies.

We appreciate this very crucial suggestion. As the reviewer says, the paper by Liu et al. (2020) nicely utilized optogenetic tagging and found of social interaction-related PV interneuronal activity. While we could not use such a sophisticated technique, we discussed how our study is related to these existing papers as follows:

In the Introduction part: “Recently, LFP oscillations in the PFC have been shown to modulate social behavior. A PFC oscillation at a low γ-range (20–50-Hz) band mediated by interneurons facilitates social interaction (Liu et al., 2020). Consistently, autism mouse models with social deficits exhibit impairments in PFC interneuronal activity (Han et al., 2012) and γ oscillations (Cao et al., 2018). These studies suggest a key role of PFC γ-range signals in the modulation of social behavior.” (Line 60-66)

In the Discussion part, we made a new paragraph: “Many previous studies have established that PV interneurons are a crucial cell type for generating cortical low γ-range (20–60 Hz) oscillations (Whittington et al., 1995; Bartos et al., 2007; Cardin et al., 2009; Sohal et al., 2009; Buzsaki and Wang, 2012; Nakamura et al., 2015; Cao et al., 2018; Liu et al., 2020). The PV-interneuron-mediated γ oscillation in the PFC is enhanced during social interaction (Liu et al., 2020) or attenuated in autism mouse models with social deficits (Cao et al., 2018), highlighting the importance of PFC PV interneurons in the expression of social behavior (Han et al., 2012; Xu et al., 2019) through the regulation of low γ oscillations (Cao et al., 2018; Liu et al., 2020). These studies utilized a protocol with 40-Hz photostimulation of PV interneurons with the aim of increasing PFC LFP power at the corresponding frequency (i.e. low γ) band (Cao et al., 2018; Liu et al., 2020). Notably, in our study, we initially sought to determine a protocol that could reduce dmPFC 4–7 Hz power to mimic the LFP power changes observed during approach behavior and eventually found that 40-Hz photostimulation selective to PV interneurons was optimal to meet this technical requirement, resulting in a decrease in dmPFC 4–7 Hz and an increase in 40 Hz power. The mechanisms underlying these reciprocal power changes are possibly mediated by a stimulation-induced interference of the entrainment of PV interneuronal spikes by a 4–7 Hz oscillation. Our study demonstrated that such optogenetic manipulations of PV interneurons extend to the BLA circuit and were sufficient to restore social interaction behavior that was reduced in social defeat stress-induced depression mouse models, similar to autism mouse models reported previously (Cao et al., 2018). Here, we note that we did not confirm that the phase-locked dmPFC interneurons observed in this study corresponded with PV-positive interneurons. To address these issues, further confirmation is necessary using techniques to identify cell types of recorded neurons such as optogenetic tagging (Liu et al., 2020).” (Line 519-544)

In addition, we discussed how our results are comparable to the results by Liu et al. (2020) (Line 480-493). For more detail, please see Reviewer #1 Major comment #6.

2. The authors show an increase in 4-7 Hz power and decrease in 30-60 Hz power during social interaction, and that these changes are associated with social avoidance behavior. While a majority of animals show a positive preference for social targets (Figure 1B), the use of CD1 animals as target animals could lead to more social avoidance, as CD1 animals are substantially larger and may increase more social avoidance behavior than regular C57BL/6J animals. The authors could use unfamiliar C57BL/6J mice as target animals. Also, the authors could incorporate an additional control with a toy mouse instead of an empty cage in the experimental setup.

We appreciate this crucial suggestion. It is also raised by Reviewer #1.

As the reviewer pointed out, we utilized a larger CD-1 mouse as a target mouse in the test, which may induce more anxiety-related or fearful behavior. To reduce the effects of these emotional factors, we performed additional tests using an unfamiliar C57BL/6J mouse with a similar body size as a target mouse (new Figure 1J). In this condition, we found significant changes in dmPFC 4–7 Hz and 30–60 Hz power in the target session, similar to the results from the larger CD-1 mouse as shown in Figure 1F. This result confirms that the dmPFC power changes are induced by social behavior while they are less associated with emotional components such as anxiety.

In addition, as a control test with no social behavior, we performed additional experiments by placing a plastic toy mouse in the cage as a novel object instead of a real mouse (new Figure 1K). In this condition, we found no significant dmPFC 4–7 Hz and 30–60 Hz power changes between the two sessions. This result confirms that the dmPFC power changes are induced specifically in a condition where mice exhibit social behavior while they are less associated with a novelty.

These results are presented in Figure 1J and 1K and described in the Results part (Line 168-184).

3. The authors show that 4-7 Hz and 30-60 Hz power is heavily modulated during specific types of social behavior, such as approaching vs. leaving the interaction zone. This suggests that the neural activity changes in Shank3 or socially defeated animals reported in Figures 2 and 3 could simply reflect changes in behavior compositions, as opposed to changes in the underlying neural encoding of these behaviors. The authors could perform more in-depth analyses of the behaviors of Shank3 or socially defeated animals using a similar approach to determine whether and how Shank3 or socially defeated animals display different compositions of behavior in their assays. The authors could also investigate the modulations of 4-7 Hz and 30-60 Hz power during these specific behaviors in Shank3 or socially defeated animals.

We appreciate this constructive suggestion.

First, in Supplementary Figure 4A, we presented additional information about the percentages of stay areas in the two sessions (left) and approach/leaving behavior in the target session (right) in the three mouse groups.

Second, similar to the wild-type mice in Figure 5, we analyzed how social approach behavior affects LFP patterns in Shank3 KO mice. While Shank3 KO mice exhibited less social interaction, they still showed substantial approach behavior (Supplementary Figure 4A). We thus applied the same analysis to Shank3 KO mice and found that they did not show significant differences in dmPFC 4–7 Hz and 30–60 Hz power between approach and leaving behavior (Supplementary Figure 4D). This result suggests that social behavior-related dmPFC activity is not properly regulated in socially deficient mice. These results are added in the Results part (Line 336-340). In defeated mice, this analysis was not applicable because most of them did not have sufficient social approach trials (Figure 3A and Supplementary Figure 4A).

Third, similar to socially deficient mice, we newly analyzed how social avoidance behavior affects LFP patterns in the wild-type mice in Figure 4A and found no significant changes in LFP power by social avoidance behavior. This analysis is now comparable to the analyses applied to Shank3 KO and defeated mice. Overall, the new Figure 4 clearly demonstrated differences across wild-type mice, Shank3 KO mice, and defeated mice.

Taken together, these mouse groups were consistently tested by the same analyses in Figure 3, 4 and Supplementary Figure 4D, demonstrating apparent differences between wild-type mice and socially deficient mice.

4. Given that the behavior repertoire is diverse across wild-type mice (Figure 1B), the current work would benefit from examining whether the level of social motivation is related to the magnitude of change of dmPFC 4-7 Hz signal in wild type mice. More specifically, this can be addressed by splitting wild-type mice into subgroups with high and low social motivation, followed by examining if the change of dmPFC 4-7 Hz band in the low social motivation subgroup is higher than that of the high social motivation subgroup, and whether it is comparable to Shank3 and socially defeated groups. Alternatively, correlating percent time of social interaction to the change magnitude of dmPFC 4-7Hz through a simple linear regression model would also help enhance the generalizability of the claim overall.

We appreciate this insightful idea. Such individual differences are also our major research interests.

Before splitting our mice into subgroups, we first directly applied correlational analyses to all mice. In Supplementary Figure 2C, we plotted the SI ratios of individual wilttype mice against their percentages of dmPFC LFP power changes in the target session. However, we could not find significant correlations between these variables. These results demonstrate that the individual differences in social behavior are not crucially related to the dmPFC power changes at least within the wild-type mouse group tested in our study. These results are now described in the Results part (Line 139-145). Therefore, we did not apply further analyses (e.g. splitting into subgroups) on this dataset.

To compare with the results from socially deficient mice, we superimposed plots obtained from the Shank3 KO and defeated mice in Supplementary Figure 2C.

5. While the authors simultaneously recorded LFP signals from both dmPFC and BLA, the temporal relationship between dmPFC and BLA oscillatory signals was not explored. The impact of the current work can be bolstered through a deeper characterization of their temporal relationship. The simultaneous LFP recording in both regions provide an opportunity to examine the phase relationship of the 4-7 Hz band (and others) during different social epochs (e.g. approaching/leaving the interaction zone) which may shed light on whether PFC-amygdalar communication is involved in social anxiety-related behaviors.

We appreciate this great idea. This computation was possible as we obtained simultaneous recordings from the dmPFC and BLA from six wild-type mice and seven Shank3 KO mice.

First, we computed dmPFC-BLA coherence and Granger causality from all no target and target sessions in Figure 1H and 1I. Similar to the power changes in Figure 1F and 1G, dmPFC-BLA coherence at the 4–7 Hz band in the target session was significantly higher than that in the no target session, confirming the coordination of dmPFC-BLA 4–7 Hz. The Granger causality spectrum exhibited a significantly higher Granger causality index at the 4–7 Hz band for the direction from the dmPFC to BLA than that for the direction from the BLA to the dmPFC, possibly representing the preferential projection of the dmPFC to the BLA (Gabbott et al., 2005; Bukalo et al., 2015). These results are added in the Results part (Line 129-138).

Second, we applied the same analysis to social avoidance behavior in Shank3 KO mice. However, dmPFC-BLA coherence and the directionality at the 4–7 Hz band were not prominent during social avoidance behavior. These results are presented in Supplementary Figure 4B and 4C and described these results at Line 252-257.

Third, we applied the same analysis to approach and leaving behavior in wild-type mice. Consistent with the LFP power changes (Figure 5F), dmPFC-BLA coherence at the 4– 7 Hz band during leaving behavior was significantly higher than that during approach behavior (Figure 5F, right). Moreover, the Granger causality index at the 4–7 Hz band in the dmPFC-BLA direction was significantly higher than that in the BLA-dmPFC direction during approach leaving behavior, but not during approach behavior (Figure 5G). These results suggest that functional information transfer at the 4–7 Hz band from the dmPFC to the BLA is lowered during social approach behavior, compared with leaving behavior. These results are added in the Results part (Line 326-335).

Reviewer #3:Here the authors are using optogenetics combined with in vivo recordings of mouse prefrontal cortex and amygdala during social behavior to test the sufficiency of different frequency bands for those social behaviors. Once establishing how the oscillations change during social behavior in wildtype mice, they identify how these oscillations are similar and different in two mouse models of neuropsychiatric disorders with decreased social interactions. In one of the disease models, they use optogenetics to enhance the power of the oscillations and find that it increases social behavior in that model.The authors examine modulation of theta and γ power in dorsal medial prefrontal cortex (dmPFC) and basolateral amygdala (BLA) during social exploration behavior. The authors performed simultaneous recordings from the dmPFC and BLA during a social interaction test and found increases in theta power and decreases in γ power compared to the non-social interaction trial. Social defeat animals and Shank3 KO mice had similar alterations in their theta frequency bands compared to control mice. Optogenetic manipulation of γ frequency bands increased social exploration behavior in social defeat mice.The strengths of the paper include the importance of the question being asked, including the brain regions studied and in studying the potential for convergent biological changes in abnormal behavior arising from different etiologies. Experimentally, a strength is the combination of simultaneous in vivo physiology of two brain regions with optogenetics in awake, behaving mice. These are difficult experiments. A further strength is the nuanced analysis of animal behavior and the measurement of brain oscillations during those phases (i.e. approach vs. leaving).The wildtype data are intriguing and interesting, as they help fill in the gaps in our understanding of how and when the prefrontal cortex and amygdala activate during social behavior, and how this is similar and different in disease models that display decreased social interactions.There is an over-reliance on multiple t-tests instead of using more appropriate statistical analyses such as ANOVA.A barrier to interpreting the results of the disease model work is the lack of appropriate controls. The authors do not use littermate controls for the Shank3 KO vs. wildtype comparisons. Instead, they compare Shank3 KO mice from one line of mice to wildtype mice from another line. It is well-known that different lines of mice can have drastically different behavior and physiology. Second, there are no untreated control animals for any of the social defeat optogenetics experiments, so it is unknown if the effects observed are specific to the disease model or nonspecific effects. There are appropriate eYFP-only controls and no-light controls.There are multiple recommendations for improvement:For all figures, I recommend showing individual data points for each animal in addition to the mean +/- SEM. There is no clear reason this is done in some graphs but not others.

These were simply our fault.

We now presented all individual data plots in addition to mean +/- SEM in all figures.

Figure S2 – I wonder if performing multiple paired t-test is the correct statistical treatment for the comparisons, since you're dividing the data into multiple different frequency bands in the same animal.

To counteract the problem of multiple comparisons, we applied Bonferroni correction in all graphs in Supplementary Figure 2. In addition, we applied this correction for multiple comparisons in all figures, where necessary (all changes were highlighted in blue in the manuscript). The statistical results were not altered from the previous manuscript.

For the data presented in Figure 1 and S2, it is unclear if the powers measured are for the whole trial or only when the animal is in the "interaction zone".

In Figure 1E-G, Figure 3, and Supplementary Figure 2, the power was computed from an entire period in the no target/target sessions.

We added an explanation in the Results part (Line 102-104): “To compute an overall tendency of LFP power changes, a Fourier transformation analysis was applied to LFP signals from each entire session.”

We added an explanation in the Methods part (Line 724-726): “the ratio of absolute power during an entire period of a target session to that during a no target session at a 4–7 Hz or 30–60 Hz band was computed.”

In figure 1E, it is unclear what is being averaged if this is a single mouse. Data are presented as the mean +/- SEM, but it’s unclear what samples are used to calculate this.

It was our mistake. In Figure 1E, the data was from all mice.

In the revised manuscript, we now presented new LFP spectrums “without normalization” in Figure 1E top. These spectrums were obtained by averaging over all wildtype mice (described at Line 104-107 and Legends). In addition, we presented all datasets from individual wild-type mice in Supplementary Figure 2A.

Consider adding a horizontal line to Figures 1 F and G showing where Z = 0. It is unclear what is being used as the baseline for the Z score.

In these figures (Figure 1F and 1G), we removed the analysis with z-scores to avoid confusion and changed them to %changes. These results are now directly comparable to the Shank3 KO and defeated mice in Figure 3B and 3C.

N=14 mice for the dmPFC recordings and N=6 for the BLA recordings. If all recordings were done with both locations targeted but fewer BLA animals are presented, please indicate why (presumably because many of the BLA recordings were not targeted correctly).

We had two reasons for this comment.

First, for the initial eight wild-type mice, we targeted only the dmPFC as we first started this study to simply understand the role of the PFC. After that, for additional six mice, we targeted both the dmPFC and BLA. We described these numbers of animals in the Methods part (Line 618-620).

Second, as the reviewer says, in some mice, some electrodes were out of the BLA while they were targeted. We described the exact numbers of animals in the Methods part (Line 711-716).

The order of data presentation in Figure 2 is confusing because it leads the reader to think that the data presented in 2C and D have something to do with the drawing presented in 2B. Based on the text, the drawing in 2B only pertains apparently to the data presented in 2E and F.

As the review suggested, this was a confusing presentation. We replaced the orders of graphs in Figure 3 and 4.

P7, L134-136 – it is unclear if this interpretation is based on the KO mice having lower social behavior than WT mice as a group, or on a per-animal basis.

This statement is based on the statistical results of inter-group comparisons between wild-type, Shank3 KO, and defeated mouse groups in Figure 3B. We re-wrote the manuscript as follows (Line 238-240): “These results suggest that the increases in dmPFC 4–7 Hz power during a target session are more prominent in Shank3 KO mice, compared with wild-type mice.”

The Shank3 KO mice appear to be tested without a WT littermate control group. The WT mice used in the initial part of the study are a separate cohort of mice from a different lineage and are then used for comparison with the Shank3 KO mice.

As the reviewer suggested, it was an exactly ideal control group. Unfortunately, the same WT littermates to Shank3 KO mice were no more available due to housing problems in our vivarium.

We added this limitation in the Methods part as follows: “The datasets obtained from Shank3 KO mice were compared with those obtained from the wild-type mice, not littermates of Shank3 KO mice.” (Line 575-577)

In addition, to avoid confusion, we combined the comparisons across WT mice, Shank3 KO mice (previous Figure 2), defeated mice (previous Figure 3) in one Figure (new Figure 3).

P7, L137-138 – it is unclear whether the Shank3 mice spending "substantial" time in the avoidance zone is different from WT mice. For the associated Figure 2EandF, there is no point of comparison with wildtype mice to know if control mice also display similar changes in theta power when in the avoidance zone.

First, we presented behavioral patterns (e.g. the percentage of social avoidance) in all mouse groups in Supplementary Figure 4A.

Second, we newly analyzed how social avoidance behavior affects LFP patterns in the wild-type mice in Figure 4A, showing no significant changes in LFP power by social avoidance behavior. This analysis is now consistent with the analyses applied to Shank3 KO and defeated mice. Overall, the new Figure 4 now clearly demonstrated differences in LFP power changes between wild-type mice and socially deficient mice.

The authors find increases in theta power during social trials and decreases in theta power during social avoidance in Shank3 KO and social defeat animals.The analysis of the spiking data is confusing. For example, p12, L248 says neurons were excluded from further analysis if they didn't show a preference in phase locking for approaching vs. leaving. Then the authors discuss how the remaining FS neurons showed higher theta phase locking during leaving. The confusing part is that then they discuss how RS neurons "did not show such significance". By definition, the neurons should have been excluded from this very analysis if they didn't show a significant difference between leaving and approaching.

In Figure 6C, we first defined significant MVLs within each behavioral pattern (approach or leaving). After this definition, the neurons that we excluded were “neurons that showed significant MVL in both approach and leaving behavior” (the neurons indicated by the red lines in Figure 5C; n = 5 RS neurons and 3 FS neurons), not “neurons that did not show a significant difference between leaving and approaching”.

As these neurons always exhibit phase locking spikes, irrespective of behavioral patterns (both approach and leaving), we did not need to test their behavior-spike relationship and thus excluded from further statistical analyses.

Even after this cell exclusion, FS neuronal populations (that did not show significant phase locking in both approach and leaving) showed significantly higher MVL for the 4–7 Hz oscillations during leaving periods, compared with approach periods, showing that a subset of FS neurons alter their entrainment to the 4–7 Hz oscillations depending on animal’s behavioral patterns.

On the other hand, RS neuronal populations (that did not show significant phase locking in both approach and leaving) did not exhibit such significant changes depending on behavioral patterns.

We modified our explanations at Line 371-381.

For Figure S4, I am concerned that doing multiple paired t-tests is not the appropriate statistics for these multiple frequency bands. I would recommend at least an ANOVA.

Thank you for this crucial comment. We applied Bonferroni correction and found no changes in our results in Supplementary Figure 6 (previous Supplementary Figure 4). Here, as the numbers of neuron samples are different across frequency bands (because behavior-irrelevant phase-locked neurons were excluded from these statistical analyses), ANOVA was not used.

In addition, we applied this statistical correction for multiple band comparisons in all figures, where necessary (they are highlighted in blue in the manuscript). The statistical results were not altered from the previous manuscript.

The figures are aesthetically pleasing.

Thank you for your compliments.